# Long-range migration of centrioles to the apical surface of the olfactory epithelium

**Kaitlin Ching[1†‡], Jennifer T Wang[1†§], Tim Stearns[1,2]\***

[1]Department of Biology, Stanford University, Stanford, United States; [2]Department of Genetics, Stanford University School of Medicine, Stanford, United States

**Abstract** Olfactory sensory neurons (OSNs) in vertebrates detect odorants using multiple cilia, which protrude from the end of the dendrite and require centrioles for their formation. In mouse olfactory epithelium, the centrioles originate in progenitor cells near the basal lamina, often 50–100 µm from the apical surface. It is unknown how centrioles traverse this distance or mature to form cilia. Using high-resolution expansion microscopy, we found that centrioles migrate together, with multiple centrioles per group and multiple groups per OSN, during dendrite outgrowth. Centrioles were found by live imaging to migrate slowly, with a maximum rate of 0.18 µm/minute. Centrioles in migrating groups were associated with microtubule nucleation factors, but acquired rootletin and appendages only in mature OSNs. The parental centriole had preexisting appendages, formed a single cilium before other centrioles, and retained its unique appendage configuration in the mature OSN. We developed an air-liquid interface explant culture system for OSNs and used it to show that centriole migration can be perturbed *ex vivo* by stabilizing microtubules. We consider these results in the context of a comprehensive model for centriole formation, migration, and maturation in this important sensory cell type.

**\*For correspondence:**
stearns@stanford.edu

[†]These authors contributed equally to this work

**Present address:** [‡]Department of Molecular, Cell, and Developmental Biology, University of California, Los Angeles, Los Angeles, United States; [§]Department of Biology, Washington University in St. Louis, St. Louis, United States

**Competing interest:** The authors declare that no competing interests exist.

## Editor's evaluation

This work will interest cell and developmental biologists studying centriole biogenesis, cilia assembly, neuron biology and development of the olfactory system. Using expansion microscopy as well as new culturing and live imaging techniques, the authors provide an unprecedented view of centriole duplication, migration, and maturation in mouse adult olfactory sensory neurons (OSNs). The major advance here is mostly technical, but the study beautifully lays the groundwork for using OSNs as a model to understand the cell biological underpinnings of organelle assembly, motility and maturation in an adult mouse tissue *in vivo*.

## Introduction

Chemosensation in many animals is mediated by ciliated neurons that have a sensory cilium at the end of a dendrite, positioned to sense the external environment. In vertebrates, olfactory sensory neurons (OSNs), found within the olfactory epithelium, have multiple cilia in a dendritic knob exposed to odorants (*Cuschieri and Bannister, 1975*; *Menco, 1997*; *McEwen et al., 2008*; *Oberland and Neuhaus, 2014*). The structure and function of these cilia are critical to olfaction. They provide a specialized membrane area for signaling via G protein-coupled olfactory receptors, and disruption of cilium structure and function can result in anosmia (*Jenkins et al., 2009*; *Kulaga et al., 2004*; *McEwen et al., 2007*). The microtubule structure of each cilium is templated by a centriole at its base, which is also referred to as a basal body. Centrioles in many cell types also recruit pericentriolar material (PCM) to form the centrosome, the major microtubule organizing center.

To create the requisite number of olfactory cilia, cells must amplify centrioles from an initial pair. We previously demonstrated that centriole amplification occurs in mitotically active progenitor cells located near the basal lamina of the olfactory epithelium in mice (*Ching and Stearns, 2020*). This is unique among cells with multiple cilia as epithelial cells with multiple motile cilia in the trachea, oviduct, and brain ependyma (multiple motile cilia cells, MMCCs) amplify centrioles only after mitotic divisions are completed.

Progenitor cells generate OSNs during embryonic development and throughout adult life, continuing to generate new ciliated olfactory neurons long after the epithelium has been established (*Moulton and Beidler, 1967*; *Mulvaney and Heist, 1971*). As a daughter cell from the progenitor division differentiates into an immature OSN, it extends projections in opposite directions: a dendrite toward the apical surface to detect odorants, and an axon through the basal lamina toward the olfactory bulb. *In vivo* fate mapping in mouse has shown that this process begins 1 day after the progenitor's last cell cycle (*Rodriguez-Gil et al., 2015*). During neurite extension, the cell body of the neuron remains close to the basal lamina (*Figure 1A*; *Rodriguez-Gil et al., 2015*).

Centrioles are amplified in the cell body and must migrate to the apical surface, traversing a distance of 50–100 µm or more (*Figure 1A*). Using transmission electron microscopy, migrating centrioles have been previously observed in mammalian OSNs (*Heist and Mulvaney, 1968*; *Mulvaney and Heist, 1971*). Within the thin, 1- to 2-µm-wide dendrites, centrioles were found to be grouped, with more groups being found in the most apical portion of the epithelium (*Mulvaney and Heist, 1971*). Six days after the progenitor cell's last cell cycle, immature OSNs differentiate into mature OSNs and express markers of maturation, such as olfactory marker protein (OMP) (*Rodriguez-Gil et al., 2015*). Upon maturation, they form multiple cilia at the apical end of the dendrite, and their nuclei migrate towards the apical surface in the days to weeks that follow (*Rodriguez-Gil et al., 2015*).

In OSNs, centrioles must be moved to the apical surface of the olfactory epithelium to assemble sensory cilia where they will have access to odorants. Many cell types across a variety of organisms are known to relocate or reorient their centrioles, including ciliating vertebrate cells (*Dawe et al., 2007*), *Drosophila* nurse cells in the germarium (*Mahowald and Strassheim, 1970*; *Bolvar et al., 2001*), and macrociliary cells in the comb jelly *Beroe* (*Tamm and Tamm, 1988*). A particularly relevant example is the *Caenorhabditis elegans* PQR neuron, in which a single centriole migrates approximately 5 µm along the dendrite by the activity of the minus-end directed microtubule motor, dynein (*Li et al., 2017*). In contrast to the PQR neuron, vertebrate OSNs each contain multiple centrioles, move those centrioles over a 10-fold greater distance, and develop throughout the life of the animal, which frequently requires centrioles to migrate through an established epithelium. Despite these biologically interesting challenges, the mechanisms of movement and the molecular composition of migrating OSN centrioles are not known.

Here, we develop new techniques for imaging and manipulating OSNs and use these to explore the mechanisms by which multiple centrioles migrate and become competent to form the multiple cilia that are important for the sensory function of OSNs. Our findings inform our understanding of OSN cell biology, as well as long-range organelle movement more broadly.

## Results

### Identification of migrating and mature centrioles in the olfactory epithelium

The olfactory epithelium is composed of multiple cell types, including OSNs and non-neuronal cell types known as sustentacular cells (*Figure 1A*). To investigate the migration of OSN centrioles, we imaged them in fixed olfactory epithelium taken from mice expressing eGFP-centrin2, a component of centrioles (*Bangs et al., 2015*). OSNs are generated from progenitors throughout development and adult life, and we used adult mice in these experiments to enable observation of centriole migration across the full depth of the mature olfactory epithelium. *En face* imaging of the apical surface allowed us to identify dendritic knobs ensheathed in sustentacular cells with borders rich in filamentous actin (*Figure 1B and C*; *Liang, 2018*). In side views of the olfactory epithelium, migrating centrioles were identified as elongated clusters of puncta, in accordance with previous reports (*Figure 1D*; *Heist and Mulvaney, 1968*; *Klimenkov et al., 2018*; *Mulvaney and Heist, 1971*; *Ying et al., 2014*).

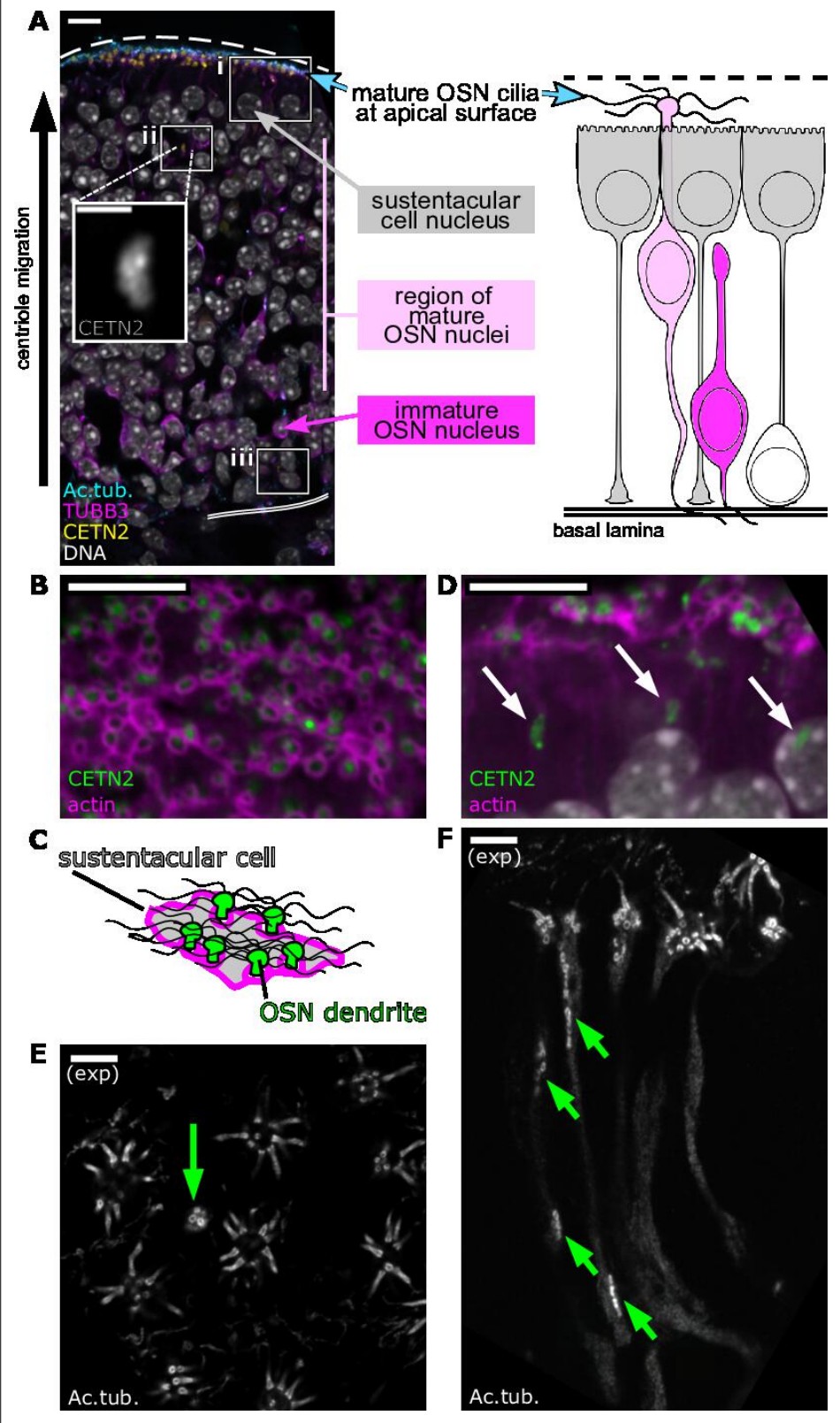

**Figure 1.** Centrioles in olfactory sensory neurons (OSNs) migrate tens of micrometers to the apical surface.
(**A**) Overview of the olfactory epithelium. Single-plane fluorescence image of a side view of the olfactory epithelium
(left), corresponding to a schematic of cell types in the olfactory epithelium (right). Yellow: eGFP-centrin; cyan:
staining for acetylated tubulin, strongly marking olfactory cilia and, faintly, neuronal microtubules; magenta:

*Figure 1 continued on next page*

*Figure 1 continued*

staining for β-tubulin III, marking neuronal microtubules; white: DAPI, marking DNA. The apical surface is oriented at the top of this image and all subsequent side-view images. Dashed line: apical surface; double solid line: basal lamina. Boxes show the relative positions of critical stages of OSN differentiation: (**i**) the subapical compartment of the olfactory epithelium, defined as the space between the bottom of the sustentacular cell nuclei and the apical surface with olfactory cilia. (**ii** and inset) A group of centrioles migrating through the middle of the olfactory epithelium, below the sustentacular cell nuclei. (**iii**) A progenitor cell near the basal lamina. Scale bar = 10 μm. Inset scale bar = 2 μm. (**B**) Mature olfactory sensory neurons. Single-plane, *en face* fluorescence image of the apical surface of the olfactory epithelium. Green: eGFP-centrin2; magenta: dye-conjugated phalloidin, marking an enrichment of F-actin at the apical borders of sustentacular cells. Scale bar = 10 μm. (**C**) Schematic of (**B**) depicting a sustentacular cell wrapping around the dendrites of nearby OSNs. Green: OSN dendrite; gray: sustentacular cell cytoplasm; magenta: F-actin; wavy black lines: multiple OSN cilia. (**D**) Centriole migration in the olfactory epithelium. Single-plane fluorescence image of a side view of the subapical compartment of the olfactory epithelium. Green: eGFP-centrin2; magenta: dye-conjugated phalloidin; arrows: groups of migrating centrioles. Scale bar = 10 μm. (**E**) Mature olfactory sensory neurons, as imaged by expansion microscopy. Single-plane fluorescence image of the *en face* apical surface. White: staining for acetylated tubulin, marking centrioles, cilia, and, faintly, neuronal microtubules. Multiple cilia can be seen protruding from mature OSNs. A green arrow marks a dendrite with a group of migrating centrioles arriving at the apical tip. Scale bar = 2 μm. (**F**) Centriole migration, as imaged by expansion microscopy. Single-plane fluorescence image of a side view of the subapical compartment. White: staining for acetylated tubulin, marking centrioles, cilia, and, faintly, neuronal microtubules. Cilia can be seen at the apical surface, and green arrows mark groups of migrating centrioles. Scale bar = 2 μm.

The online version of this article includes the following figure supplement(s) for figure 1:

**Figure supplement 1.** Centriole and cilium numbers in olfactory sensory neuron (OSN) progenitors and mature OSNs.

To visualize centriole migration in OSNs at higher resolution, we applied expansion microscopy (*Gambarotto et al., 2019*; *Sahabandu et al., 2019*) to sections of adult olfactory epithelium. We were able to reproducibly expand these samples approximately four-fold and could resolve olfactory cilia in mature OSNs, as well as migrating centrioles in immature OSNs (*Figure 1E and F*) by staining for acetylated tubulin, which marks centrioles, cilia, and, more faintly, the microtubules of the dendrite. *En face* imaging of the dendritic knob using expansion microscopy identified groups of centrioles at the apical surface (*Figure 1E*). Side views of the olfactory epithelium identified groups of migrating centrioles positioned below the apical surface (*Figure 1F*).

With the increased resolution afforded by expansion microscopy, we counted centriole number in progenitor cells and mature OSNs by staining for acetylated tubulin and using a semi-automated volume segmentation and counting pipeline (*Figure 1—figure supplement 1A and B*). We found that basally positioned progenitor cells with amplified centrioles harbored, on average, approximately 26 centrioles (SEM = 2.234; standard deviation = 9.992; n = 20 cells from two mice, *Figure 1—figure supplement 1C*). In mature OSNs that had completed centriole migration with centrioles docked at the dendritic knob, each cell had, on average, approximately 36 centrioles (SEM = 1.665; standard deviation = 8.156, n = 24 dendritic knobs from two mice, *Figure 1—figure supplement 1C*). These results suggest that progenitor cells form the majority of centrioles present in the dendritic knob of mature OSNs, in keeping with our previously published work (*Ching and Stearns, 2020*).

To determine whether all of the centrioles at mature dendritic knobs nucleate cilia, we counted the number of cilia found in each dendritic knob (*Figure 1—figure supplement 1B and D*). We found that, on average, 85% of centrioles at a dendritic knob were associated with cilia (SEM = 2.47; standard deviation = 12.10), with an average of approximately 31 cilia per knob (SEM = 1.455; standard deviation = 7.126). It is unclear whether the 15% of centrioles not associated with a cilium previously had a cilium, will eventually make a cilium, or whether they are excess to an independent process that determines cilium number.

## Groups of OSN centrioles migrate from the basal lamina to the apical surface in tandem with dendrite outgrowth

The dendrites of mature OSNs extend from the cell body in the basal region of the olfactory epithelium to the ciliated dendritic knobs at the apical surface. In our expansion microscopy images, we noticed that centrioles within dendrites were often grouped together (*Figure 1F*). We first tested

whether multiple groups of centrioles could be present within a single dendrite, as suggested by previous TEM results (*Heist and Mulvaney, 1968*; *Mulvaney and Heist, 1971*). We imaged immature OSNs using expansion microscopy through the entire thickness of the dendrite. We found that centrioles were positioned in multiple groups within the same dendrite (*Figure 2A*).

We next asked whether the dendrite completely extends prior to centriole migration or whether migration occurs in tandem with dendrite extension. We imaged growth-associated protein 43 (GAP43), a marker of immature OSNs and some progenitor cells, to observe the extent of dendrites during outgrowth (*McIntyre et al., 2010*). We observed centrioles in partially elongated dendrites, indicating that the dendrite does not fully extend prior to centriole migration (*Figure 2B*). The leading portions of the dendrites had a variety of growth cone morphologies, and centriole groups lagged behind the end of the dendrite by as much as 8.4 µm (*Figure 2B*, mean = 3.68 µm, SEM = 0.80 µm, standard deviation = 2.76 µm, n = 12 dendrites, N = 2 animals). By probing for β-tubulin III (TUBB3), a tubulin isotype expressed primarily in neurons, we observed that microtubules were present both apical to and basal to the migrating centrioles (*Figure 2B*). These data indicate that the leading group of centrioles migrates in tandem with dendrite extension and can be followed by lagging groups within the dendrite.

We established an *ex vivo* live imaging system for dissected olfactory epithelium to begin to assess the mechanisms by which centrioles migrate. To measure rates of centriole movement in the final stage of migration, we used time-lapse confocal microscopy to image migrating eGFP-centrin2-labeled centrioles. Flattened olfactory epithelium was mounted on agarose pads and imaged *en face* immediately after dissection. Z-stacks were acquired to track the movement of centriole groups in the apical 15–25 µm of the epithelium. We found that the cells in the excised olfactory epithelium remained intact, with normal nuclear morphology and few delaminating cells over the time course of imaging. Considering net movement over the course of the sequences, centriole groups moved slowly, with the fastest moving at approximately 0.18 µm/min (n = 22 centriole groups, N = 3 animals, *Figure 2C*, *Figure 2—video 1*). Most centriole groups exhibited net movement in the apical direction (*Figure 2C*, *Figure 2—video 1*, n = 14 out of 22 groups observed), although approximately one-third had either net movement in the basal direction or no movement during the imaging period (*Figure 2D*, *Figure 2—video 2*, n = 8 out of 22 groups observed).

Our results show that centrioles often migrate in groups, moving relatively slowly towards the apical end of the dendrite. How might these groups of centrioles remain together during migration? Centrioles in dividing cells are kept in close proximity by two forms of linkage (*Breslow and Holland, 2019*; *Nigg and Stearns, 2011*). Newly formed centrioles are attached to their associated mother centriole by an engagement linker, which is broken by passage through mitosis, and disengaged centrioles are linked by cohesion fibers. Our inability to resolve the centriole-centriole relationships that define engagement in the migrating groups due to tight packing within the narrow dendrite prevents us from being able to determine whether the engagement link is responsible for their grouping. However, we consider it unlikely that all centrioles are held together this way, based on our previous results showing that centriole amplification is initiated prior to mitotic divisions in progenitors (*Ching and Stearns, 2020*).

Given that centriole engagement is unlikely to be the major determinant of grouping, we tested whether grouping is mediated by centriole cohesion. Cohesion fibers contain rootletin (CROCC) (*Vlijm et al., 2018*), which is also a component of the striated rootlet that anchors centrioles bearing cilia (*Yang et al., 2002*). We performed expansion microscopy to detect rootletin on centrioles. Strikingly, migrating centrioles lacked rootletin signal (*Figure 3A*), indicating that these groups are not held together by rootletin-based cohesion fibers. We also observed that migrating centrioles not only move in multiple groups, but that some centrioles are not grouped at all, localizing singly between migrating groups in immature OSNs (*Figure 3A*). The rootletin antibody did detect rootlets associated with centrioles in mature OSNs (*Figure 3A*), as well as cohesion fibers in tissue culture cells (*Figure 3—figure supplement 1*; *Menco et al., 1978*; *McClintock et al., 2008*). Thus, rootletin is not present in migrating centrioles and therefore centriole cohesion is unlikely to play a role in grouping centrioles together during migration.

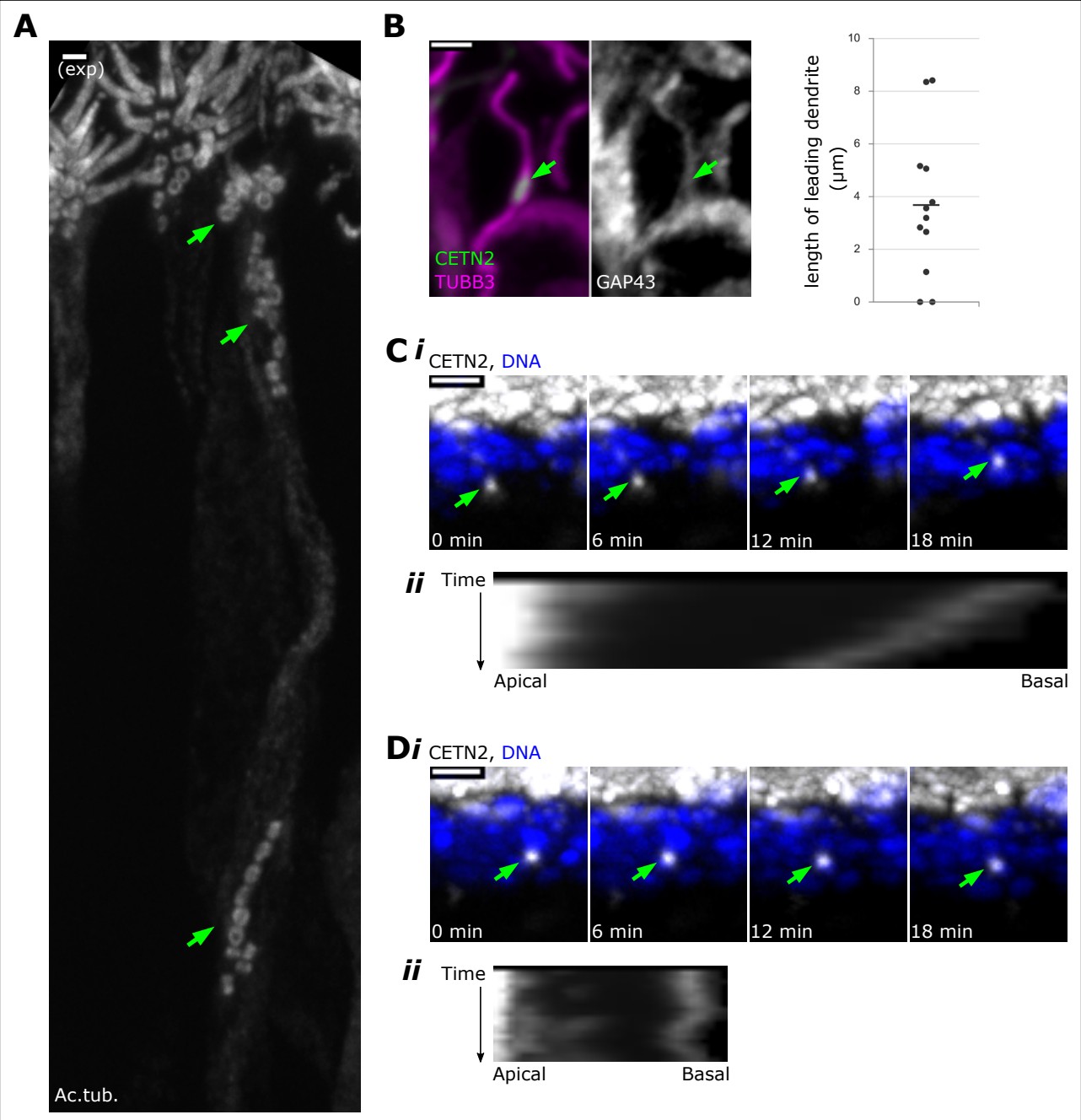

**Figure 2.** Centrioles migrate in multiple groups during dendrite elongation. (**A**) Within a single dendrite, centrioles migrate in multiple groups. Expansion microscopy – side view of expanded olfactory epithelium. In all side-view images, apical is oriented toward the top of the image. White: staining for acetylated tubulin. Maximum z-projection of confocal stack. Mature olfactory sensory neurons (OSNs) have multiple cilia, which are visible at the apical surface. Green arrows: groups of centrioles migrating separately within the dendrite of a single OSN. Scale bar = 2 μm. (**B**) Centriole migration occurs concomitantly with dendrite elongation. Fluorescence image of a side view of the olfactory epithelium, maximum z-projection of confocal stack. In the leftmost image, magenta: staining for β-tubulin III; green: eGFP-centrin2. Middle image: staining for GAP43 in the same cell; arrows: a group of migrating centrioles. Scale bar = 2 μm. Rightmost image: a plot of lengths reflecting the distance from the centriole group to the end of the dendrite. (**C**) A centriole group migrates toward the apical surface. (**i**) Live time-lapse imaging of olfactory epithelium. Maximum x-projection image, showing a side view. White: eGFP-centrin2; blue: Hoechst, marking DNA in the most apical layer of nuclei, which are mostly sustentacular cells; green arrows: a group of centrioles moving toward the apical surface at 0.18 μm/min (fastest rate of all observed groups). See *Figure 2—source data 1* for migration rates. See *Figure 2—video 1* for original video. (**ii**) Kymograph illustrating migration of the centriole group. Apical and basal direction labels indicate orientation of the sample in the kymograph. (**D**) A centriole group with no net movement. (**i**) Live time-lapse imaging of olfactory epithelium, highlighting a different centriole group from the same acquisition as (**C**). Maximum x-projection image, showing a side view. White: eGFP-centrin2; blue:

*Figure 2 continued on next page*

*Figure 2 continued*

Hoechst; green arrows: a group of centrioles that have no net movement. See *Figure 2—source data 1* for migration rates. See *Figure 2—video 2* for original video. (**ii**) Kymograph illustrating a lack of total migration of the centriole group. Apical and basal direction labels indicate orientation of the sample in the kymograph.

The online version of this article includes the following video and source data for figure 2:

**Source data 1.** Migration rates of centriole groups from time-lapse imaging.

**Figure 2—video 1.** A centriole group migrates toward the apical surface.

https://elifesciences.org/articles/74399/figures#fig2video1

**Figure 2—video 2.** A centriole group with no net movement.

https://elifesciences.org/articles/74399/figures#fig2video2

## Centriole composition shifts toward maturation during OSN development

In the typical animal cell cycle, centrioles require two cell cycles after their formation to become competent to form cilia (*Breslow and Holland, 2019*; *Nigg and Stearns, 2011*). During this maturation process, they lose proteins associated with the early steps of centriole formation, including components of the cartwheel, and acquire proteins associated with microtubule nucleation, as well as distal and subdistal appendages. To understand the progression of centriole maturation in the olfactory epithelium, we examined proteins associated with centriole formation and microtubule nucleation in progenitors, immature OSNs, and mature OSNs.

First, we examined the procentriole markers STIL (SCL/Tal1 interrupting locus protein) and SASS6 (spindle assembly abnormal protein 6) using expansion microscopy. In cycling mammalian cells grown in culture, these proteins are present during procentriole formation in S-phase and are lost from centrioles in mitosis following centriole disengagement (*Arquint and Nigg, 2014*; *Strnad et al., 2007*). In the olfactory epithelium, both STIL and SASS6 were present on amplifying centrioles in progenitors and localized to the proximal end of newly formed centrioles, supporting the model that these proteins are involved in centriole amplification from rosettes (*Figure 3B and C*). Remarkably, STIL and SASS6 are retained on at least some of the migrating centrioles in immature OSNs and are associated with the presumptive proximal end of these centrioles (*Figure 3B and C*). In mature OSNs, STIL and SASS6 are absent from centrioles (*Figure 3B and C*). Together, these results suggest that immature procentrioles are capable of migrating, while the dendritic knob is primarily formed of mature centrioles that lack STIL and SASS6.

In many differentiated neurons, centrioles lose the microtubule nucleating capability that allows them to function as centrosomes. PCM proteins involved in microtubule organization, including gamma-tubulin, CDK5RAP2 (CDK5 regulatory subunit-associated protein 2), and pericentrin (PCNT), are reduced at centrosomes in these neurons and relocalized to non-centrosomal sites (*Leask et al., 1997*; *Stiess et al., 2010*; *Yonezawa et al., 2015*; *Sánchez-Huertas et al., 2016*; *Wilkes and Moore, 2020*). We were interested in assessing the association of PCM proteins with centrioles in OSNs. We first probed for the microtubule nucleator gamma-tubulin using expansion microscopy. In both progenitor cells with amplifying centrioles and in immature OSNs, gamma-tubulin localizes to a diffuse cloud around centriole barrels (*Figure 3D*). In mature, fully ciliated OSNs, gamma-tubulin was also associated with centrioles. Next, we determined the localization of CDK5RAP2 and pericentrin, PCM proteins involved in recruiting and activating gamma-tubulin (*Fong et al., 2008*; *Choi et al., 2010*; *Delaval and Doxsey, 2010*; *Figure 3E and F*). Both CDK5RAP2 and pericentrin are present in progenitor cells with amplifying centrioles, as well as migrating centrioles in immature OSNs. In dendritic knobs, CDK5RAP2 was present, while pericentrin signal decreased. Our data show that groups of migrating OSN centrioles remain associated with microtubule nucleation proteins and PCM, in contrast to the centrioles of many other types of mammalian neurons. This is similar to MMCCs, where amplified centrioles were also reported to have associated PCM (*Jurczyk et al., 2004*; *Vladar et al., 2012*; *Zhao et al., 2021*).

## Immature OSNs form a single cilium from the parental centriole

We next used expansion microscopy to examine the events associated with arrival of the centrioles at the dendritic knob and formation of cilia. We observed that in some OSN dendrite tips with multiple

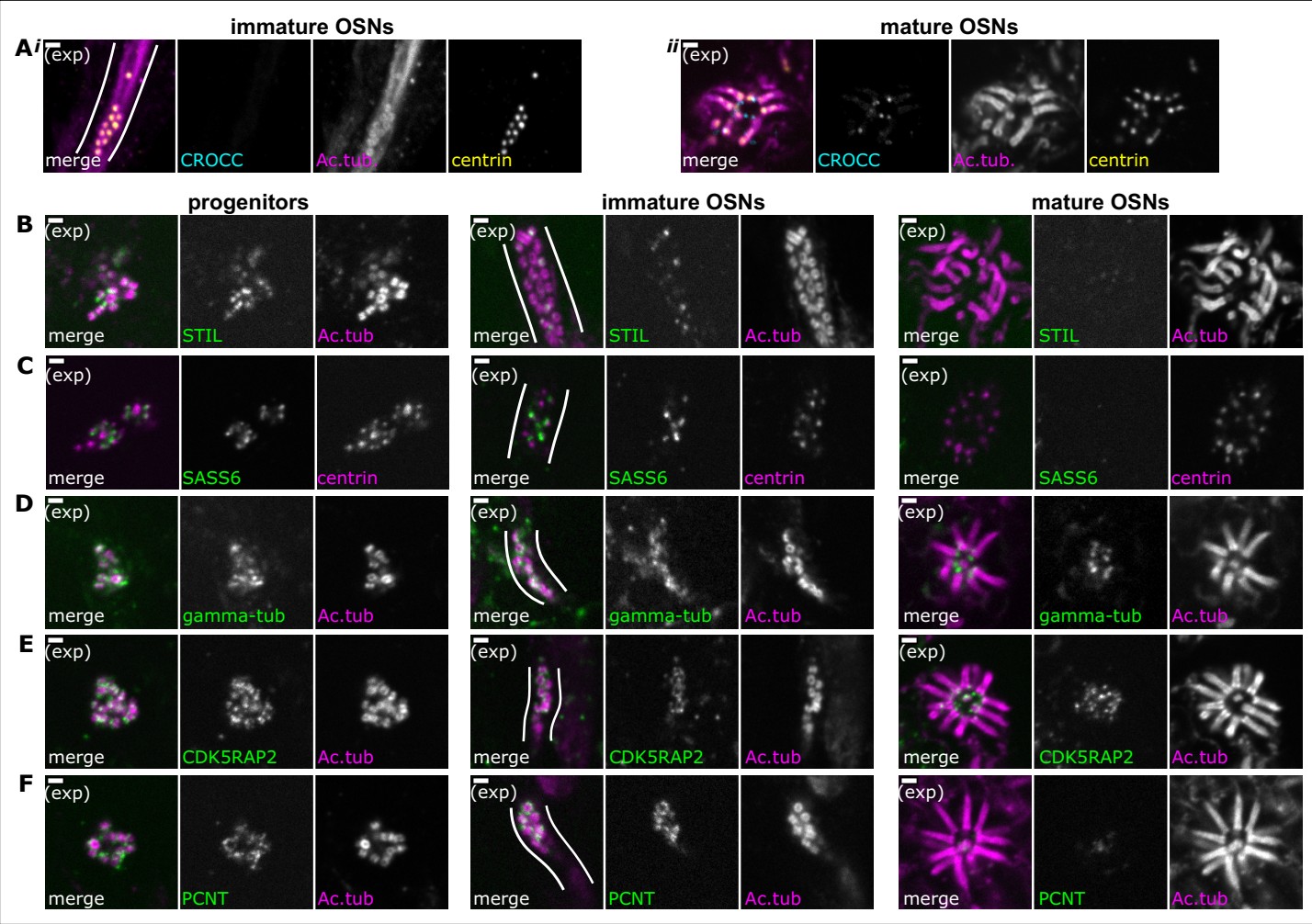

**Figure 3.** Composition of centriole groups during olfactory sensory neuron (OSN) differentiation. (**A**) The cohesion fiber and striated rootlet protein rootletin is absent during centriole migration but is gained at the mature dendritic knob. Expansion microscopy, maximum z-projection of confocal stack. In all side-view images, apical is oriented toward the top of the image. Cyan: staining for Rootletin (CROCC). Magenta: staining for acetylated tubulin; yellow: staining for centrin. (**i**) Side view of migrating centrioles. Centrioles below the apical surface, in the dendrite of an immature OSN. (**ii**) Centrioles at the apical surface in the same sample as those shown in (**Ai**). Scale bars = 2 μm. (**B–F**) Expansion microscopy – single-plane fluorescence images of centrioles in progenitor cells (left column), immature OSNs with migrating centrioles (middle column, white lines outline an OSN dendrite), and mature OSNs imaged *en face* (right column). Scale bars = 2 μm. (**B**) The immature centriole protein STIL is present in progenitors and immature OSNs. Green: staining for STIL; magenta: staining for acetylated tubulin. (**C**) The immature centriole protein SASS6 is present in progenitors and immature OSNs. Green: staining for SASS6; magenta: staining for centrin. (**D**) The pericentriolar material protein gamma-tubulin is present throughout OSN differentiation. Green: staining for gamma-tubulin; magenta: staining for acetylated tubulin. (**E**) The pericentriolar material protein CDK5RAP2 is present throughout OSN differentiation. Green: staining for CDK5RAP2; magenta: staining for acetylated tubulin. (**F**) The pericentriolar material protein pericentrin (PCNT) is present in progenitors and immature OSNs. Green: staining for pericentrin (PCNT); magenta: staining for acetylated tubulin.

The online version of this article includes the following figure supplement(s) for figure 3:

**Figure supplement 1.** Expansion microscopy staining of rootletin in cycling cells.

centrioles, near the apical surface, a single cilium was present (*Figure 4A and A'*). We supposed that this reflected an inherent asymmetry in centriole states, such that only one centriole is competent to form a cilium at this stage prior to formation of the full complement of cilia. Ciliogenesis is a multistep process that, in many organisms, requires a mature centriole to bear distal and subdistal appendages. To assess distal appendage acquisition, we used expansion microscopy to image CEP164 (centrosomal protein 164), a distal appendage protein required for ciliogenesis (*Tanos et al., 2013*; *Cajanek and Nigg, 2014*). In OSNs with a single cilium, distal appendages were present on the centriole at the base of the cilium and not at other centrioles (*Figure 4A and A'*), confirming that an inherent

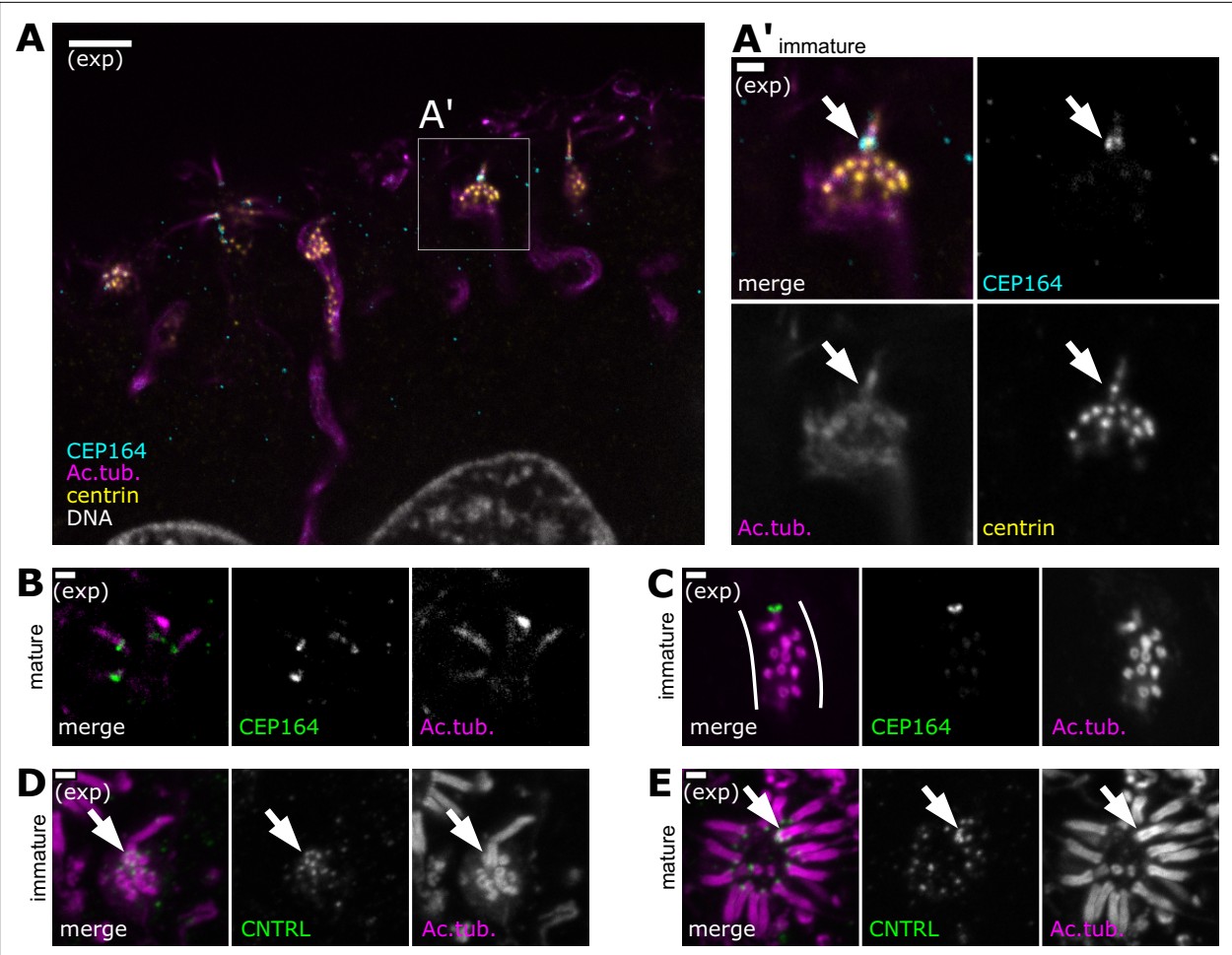

**Figure 4.** A single cilium forms prior to formation of multiple cilia. (**A**) An olfactory sensory neuron (OSN) bearing a single cilium in the subapical compartment of the olfactory epithelium. Expansion microscopy – single-plane fluorescence image of a side view of the subapical compartment. In all side-view images, apical is oriented toward the top of the image. Cyan: staining for CEP164, marking the location of distal appendages; magenta: staining for acetylated tubulin, marking centrioles, cilia, and, faintly, neuron microtubules; yellow: staining for centrin, marking centrioles; white: DAPI, marking DNA. A box marks the location of the inset (**A′**). Scale bar = 10 μm. (**A′**) Immature OSN with a single cilium, inset from (**A**). Magenta: staining for acetylated tubulin; yellow: staining for centrin; cyan: staining for CEP164, marking distal appendages at the base of the cilium; arrow: Cep164 is only found at the base of the cilium, and not present on other centrioles within the dendrite. Scale bar = 2 μm. (**B**) During centriole migration in immature OSNs, a single centriole bears Cep164. Expansion microscopy – single-plane fluorescence image of migrating centrioles in a side view of the olfactory epithelium. Green: staining for CEP164; magenta: staining for acetylated tubulin. Scale bar = 2 μm. (**C**) In mature OSNs, multiple cilia bear Cep164. Expansion microscopy – single-plane fluorescence image of a mature OSN with multiple cilia. Green: staining for CEP164, marking distal appendages at the base of cilia; magenta: staining for acetylated tubulin. Scale bar = 2 μm. (**D**) In OSNs bearing a single cilium, the centriole at the base of the cilium is surrounded by the subdistal appendage/basal foot marker centriolin. Expansion microscopy – maximum z-projection of a confocal stack. Magenta: staining for acetylated tubulin; green: staining for centriolin, marking subdistal appendages on the mother centriole at the base of the cilium. Scale bar = 2 μm. (**E**) In mature OSNs, a single cilium is surrounded by centriolin. Expansion microscopy – mature OSN imaged *en face*, maximum z-projection of a confocal stack. Magenta: staining for acetylated tubulin; green: staining for centriolin; arrow: centriolin surrounds the base of one cilium. Other cilia of the same dendrite are only associated with one centriolin punctum. Scale bar = 2 μm.

The online version of this article includes the following figure supplement(s) for figure 4:

**Figure supplement 1.** Centriole maturation in progenitors and migrating centriole groups.

asymmetry in centriole states exists. By contrast, in OSNs with multiple cilia, many centrioles acquired distal appendages, indicating that centriole maturation and remodeling occurred at the apical surface (*Figure 4B*).

How might this centriole asymmetry arise? Based on our previous results regarding centriole amplification (*Ching and Stearns, 2020*), we hypothesized that in progenitor cells a single, mature, parental centriole segregates to each daughter cell, whereas the newly amplified centrioles remain immature

and do not acquire appendages. In support of this hypothesis, we found that both in migrating centrioles (*Figure 4C*, *Figure 4—figure supplement 1A and B*) and in centrioles in the process of amplification (*Figure 4—figure supplement 1C*), only a single centriole was found to have distal and subdistal appendages. These data suggest that the centriole modifications required to make cilia occur after centriole migration during OSN differentiation, and that the already-modified parental centriole is able to make a cilium before all other centrioles. The temporal progression of centriole maturation is consistent with the presence of markers of immature centrioles on some migrating centrioles (*Figure 3B and C*).

In tracheal MMCCs, the parental centriole retains its unique structural characteristics even in mature cells after all cilia form (*Liu et al., 2020*). In particular, the parental centriole has a radial array of subdistal appendages, in contrast to all other centrioles, which bear a single basal foot. Radial subdistal appendages and basal feet share molecular components, but can be distinguished by the arrangement of these components around centrioles. To test whether the parental centriole could be distinguished from the other centrioles in mature OSNs, we stained for the shared component centriolin using expansion microscopy. In migrating centrioles, centriolin was present on only one centriole, surrounding that centriole in a subdistal appendage-like pattern (*Figure 4—figure supplement 1B*). In cells in which this centriole formed a single cilium, centriolin retained this subdistal appendage-like pattern around the base of the cilium, while other centrioles had acquired single puncta of centriolin (*Figure 4D*). In mature OSNs with multiple cilia, one centriole was surrounded by centriolin, whereas the other centrioles still had only single foci of centriolin, consistent with basal feet as observed in multiciliated epithelial cells (*Figure 4E*, *Figure 4—figure supplement 1D*). We conclude that in OSNs, as in multiciliated epithelial cells, the cilium nucleated by the parental centriole has unique structural characteristics compared to the other cilia at the mature dendritic knob.

## Microtubule dynamics are necessary for centriole migration in OSNs

Microtubules are a critical component of the neuron cytoskeleton, involved in neurite structure and transport. Neurons contain many stable microtubules as well as a population of dynamic microtubules (*Baas et al., 2016*). We sought to determine whether perturbation of the dynamic microtubules would affect centriole migration to the dendrite tip. To facilitate these experiments, we developed an explant culture system that retained the architecture of the epithelium *in vivo*. Olfactory epithelium dissected from the septa of mice expressing eGFP-centrin2 and Arl13b-mCherry was plated on transwell filters. To mimic the environment that the olfactory epithelium experiences *in vivo*, the bottom compartment was filled with medium to create an air-liquid interface at the filter (*Figure 5A*).

First, we tested whether explant cultures were viable over the required time course and whether cells retained the ability to progress through the cell cycle. We found that EdU, which labels newly replicated DNA, incorporated into nuclei near the basal lamina in the explant, indicating that cells were able to replicate DNA (*Figure 5B and C*). In addition, many of the EdU-positive cells appeared to have amplified centrioles, in accordance with our previous observations (*Ching and Stearns, 2020*). We treated explants with EdU at two time points, 1 and 6 hr post-dissection, and found that the number of EdU-positive nuclei increased from 0.63 nuclei per 100 μm basal length of epithelium to 0.96 nuclei per 100 μm of epithelium (total lengths quantitated = 634.84 μm and 621.52 μm for 1 and 6 hr, respectively). Treatment during the time course did not affect overall morphology of the olfactory epithelium. These results demonstrate that cells in the explant cultures can synthesize DNA and amplify centrioles, and that the culture retains its characteristic architecture during the time course. This early step in OSN development, in combination with our live *ex vivo* imaging of the end of centriole migration (*Figure 2C and D*), suggests that many aspects of OSN differentiation continue to occur in the hours after olfactory epithelium is removed from the animal.

Next, we used this explant culture system to assess the role of microtubules and microtubule dynamics in centriole migration. To characterize migration, we used a scoring method derived in part from *Mulvaney and Heist, 1971*, in which the authors use the stereotyped positioning of the sustentacular cells to define centriole position in OSNs. We defined the subapical compartment (sustentacular cells and OSN dendrites in final stage of extension) as the space between the bottom of the sustentacular cell nuclei, and the middle and basal compartment (progenitors and OSN dendrites in early stage of extension) as the space between the actin-rich apical domain and the bottom of the sustentacular cell nuclei and the basal lamina (*Figure 5—figure supplement 1A*). In observing

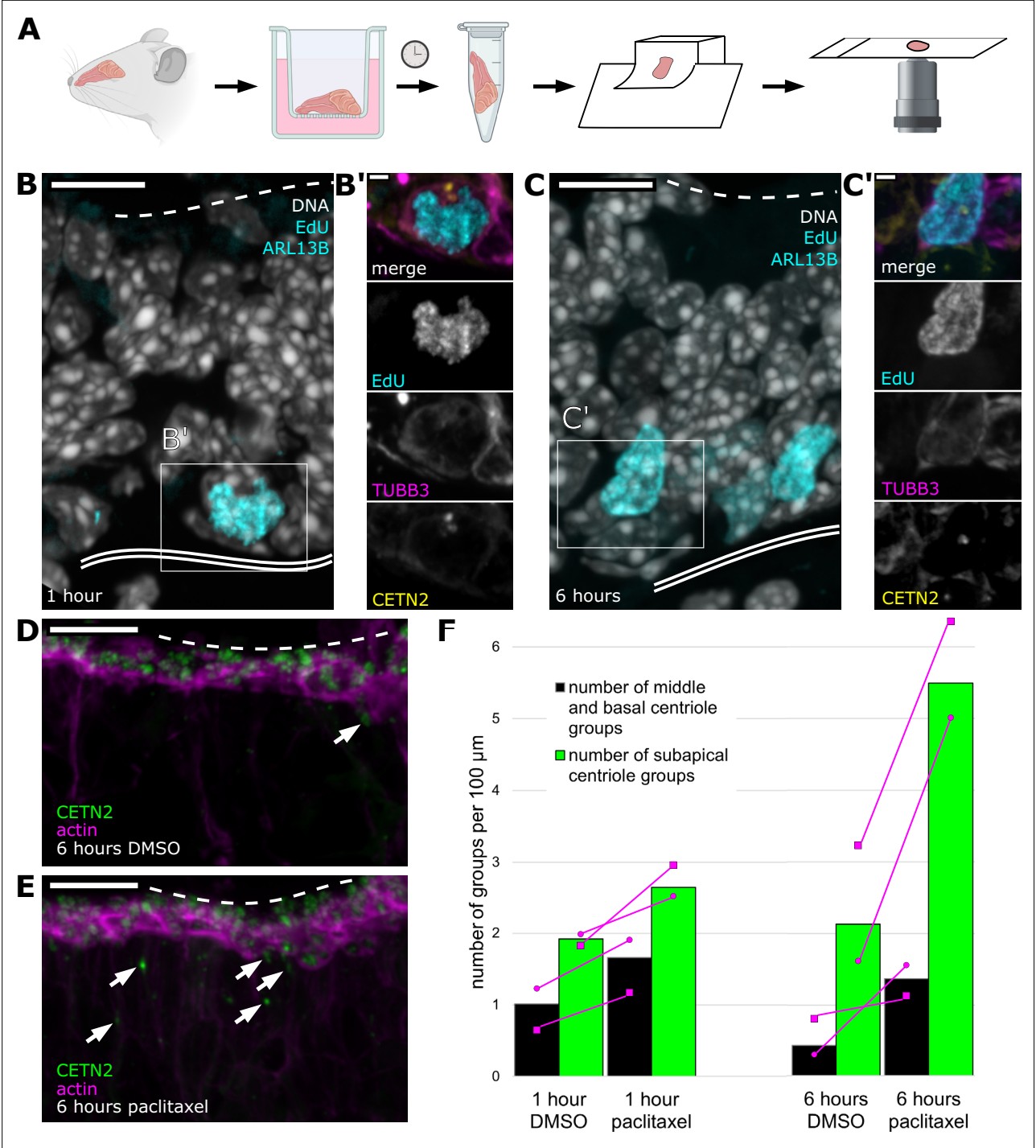

**Figure 5.** Centriole progression toward the apical surface can be altered by stabilizing microtubules. (**A**) Schematic: workflow from sample collection through imaging of olfactory epithelium explants. Septal olfactory epithelium was taken from adult mice expressing eGFP-centrin2 and Arl13b-mCherry and plated on transwell filters with an air-liquid interface. After 1 or 6 hr, samples were fixed and processed for sectioning. Stained sections were analyzed by confocal microscopy. Figure created with BioRender.com. (**B**) Progenitor cells synthesize DNA by 1 hr *ex vivo*. Fluorescence image of an olfactory epithelium explant grown in EdU for 1 hr, maximum intensity projection. Gray: DAPI, marking all nuclei; cyan: EdU, conjugated to dye after fixation, marks cells that synthesized DNA *ex vivo*; dashed line: the apical surface; double solid line: basal lamina; box: location of the inset (**B′**). Scale bar = 10 µm. (**B′**) Inset from (**B**) showing a progenitor cell positive for EdU. Magenta: staining for β-tubulin III, showing that the cell is neuronally fated; yellow: eGFP-centrin 2, showing signal consistent with centriole amplification; cyan: EdU. Scale bar = 2 µm. (**C**) More cells synthesize DNA by 6 hr *ex vivo*. Fluorescence image of an olfactory epithelium explant grown in EdU for 6 hr, maximum intensity projection. Tissue was taken from the same animal as that shown in (**B**). Gray: DAPI, marking all nuclei; cyan: EdU, conjugated to dye after fixation, shows an increased number of cells that

*Figure 5 continued on next page*

*Figure 5 continued*

have synthesized DNA *ex vivo*, compared to 1 hr treatment. Box: location of the inset (**C'**). Scale bar = 10 µm. (**C'**) Inset from (**C**) showing a progenitor cell positive for EdU. Magenta: staining for β-tubulin III, showing that the cell is neuronally fated; yellow: eGFP-centrin 2 shows signal consistent with centriole amplification; cyan: EdU. Scale bar = 2 µm. (**D**) Control image of centriole group position in explants treated with DMSO for 6 hr. Single-plane fluorescence image. Green: eGFP-centrin2; magenta: dye-conjugated phalloidin; dashed line: apical surface; arrows: migrating centriole groups. Scale bar = 10 µm. (**E**) Centriole group position in explants treated with paclitaxel for 6 hr. Single-plane fluorescence image. Tissue was taken from the same animal as that shown in (**D**). Green: eGFP-centrin 2; magenta: dye-conjugated phalloidin; arrows: migrating centriole groups. Compared to (**D**), many centrioles are found below the apical surface. Dashed line: apical surface. Scale bar = 10 µm. (**F**) Bar plot summarizing centriole group position in explants grown for 1 or 6 hr in the presence of DMSO or paclitaxel. Green bars: migrating centriole groups in the subapical compartment of the epithelium (apical surface through sustentacular cell nuclei); black bars: centriole groups in the middle and basal regions of the epithelium (between sustentacular cell nuclei and basal lamina); magenta circles: normalized number of groups for the female sample; magenta squares: normalized number of groups for the male sample. Magenta lines connect paired samples. Counts were normalized to the lateral length of the basal lamina. Length of epithelium scored for each time point: 1 hr DMSO = 988.17 µm, 1 hr paclitaxel = 905.37 µm, 6 hr DMSO = 1126.44 µm, 6 hr paclitaxel = 1019.76 µm. Number of centriole groups counted: 1 hr DMSO = 29, 1 hr paclitaxel = 39, 6 hr DMSO = 29, 6 hr paclitaxel = 70. N = 2 animals. See *Figure 5—source data 1* for values.

The online version of this article includes the following source data and figure supplement(s) for figure 5:

**Source data 1.** Counts of centriole groups - paclitaxel treatment.

**Figure supplement 1.** Migrating centriole groups in olfactory epithelium explants.

**Figure supplement 1—source data 1.** Counts of centriole groups - explants prior to treatment.

**Figure supplement 1—source data 2.** Counts of centriole groups - nocodazole treatment.

samples fixed prior to placement on filters, we noticed a high degree of variability in the number of migrating centriole groups between samples (*Figure 5—figure supplement 1B*), though all samples contained more centriole groups were found in the subapical compartment than in the middle and basal region, consistent with previous reports (*Mulvaney and Heist, 1971*). To prevent this variability from obscuring migration defects, we analyzed paired samples in which one half of the septum received drug treatment and the other half from the same animal was treated with DMSO as a control. Olfactory epithelium treated with 10 µg/mL nocodazole to depolymerize microtubules showed only a subtle change in the number of migrating centriole groups (*Figure 5—figure supplement 1C*). This may reflect that in OSNs, as is the case in many types of neurons, a portion of the microtubules are stable, even under depolymerizing conditions (*Baas et al., 2016*).

To test if the dynamic nature of microtubules is necessary for normal migration of centriole groups, we treated explant cultures for 1 hr or 6 hr with paclitaxel (taxol), a microtubule stabilizing drug, or DMSO as a vehicle control. Treatment with DMSO or paclitaxel for 1 hr or 6 hr did not affect overall tissue morphology, including the actin-rich apical domain (*Figure 5D and E*). In all conditions, more centriole groups were found in the subapical compartment than in the middle and basal region, consistent with previous reports (*Mulvaney and Heist, 1971*), and suggesting that the explant cultures recapitulate this aspect of *in vivo* development. We scored the number of migrating centriole groups in distinct regions of the olfactory epithelium with paclitaxel treatment. In sum, we found that upon treatment with paclitaxel, more centriole groups were found in the subapical and middle/basal compartments, likely in transit (p=0.0084 by paired permutation test, *Figure 5F*). To resolve the basis for this effect, we compared the 1 hr and 6 hr treatment samples. At the 1 hr time point, explants treated with paclitaxel had only slightly more (1.38-fold) centriole groups in the subapical compartment than control (N = 2 animals, n = 39 and 29 centriole groups, total basal lengths quantitated = 905.37 µm and 988.17 µm for paclitaxel and DMSO, respectively). In contrast, after 6 hr of paclitaxel treatment, the subapical compartment had 2.58-fold more centriole groups compared to control (N = 2 animals, n = 70 and 29 centriole groups, total basal lengths quantitated = 1019.76 µm and 1126.44 µm for paclitaxel and DMSO, respectively, *Figure 5F*). The accumulation of centriole groups with paclitaxel treatment, particularly in the subapical compartment, suggests that microtubule dynamicity is required for centriole groups to complete their progression to the apical membrane.

## Discussion

OSNs in vertebrates have a morphology specialized for their role, with an array of sensory cilia at the tip of a dendrite. We showed previously that the centrioles required to form these cilia are produced

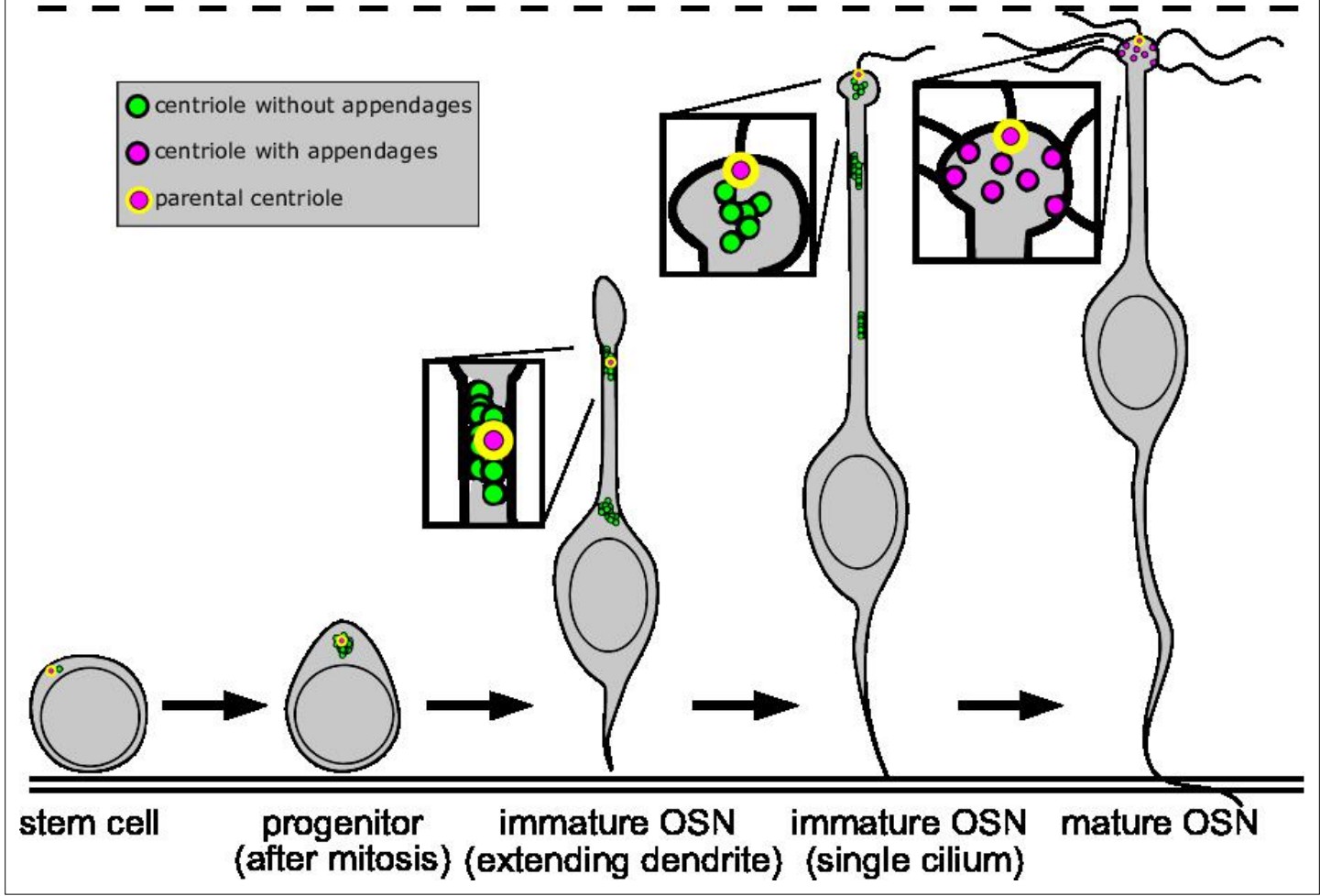

**Figure 6.** Summary of centriole migration and maturation during olfactory sensory neuron (OSN) differentiation. Diagram summarizing the migration and maturation of centrioles over the course of OSN differentiation. A double solid line marks the basal lamina, and a dashed line is drawn above the apical surface. Newly formed centrioles lacking appendages are represented as green dots, centrioles with appendages as magenta dots, and the parental centriole as a magenta dot with a yellow outline. After centriole amplification and mitosis in the progenitor, the immature OSN extends a dendrite toward the apical surface. Centrioles migrate primarily in groups during dendrite elongation. At the apical surface, a single cilium forms from the appendage-bearing, parental centriole, before multiple cilia form in the mature OSN.

in progenitor cells that undergo centriole amplification and divide before becoming OSNs (*Ching and Stearns, 2020*). Here, we have examined centriole migration and maturation in OSNs, with a focus on the unique biology of OSNs. We established new techniques for manipulating and studying the olfactory epithelium, applying expansion microscopy, live imaging, and an explant system. These will be useful for future experiments on the cells of this tissue, important for their sensory role and as a point of entry for respiratory infection.

Our results are summarized in *Figure 6*. Progenitor cells, located near the basal lamina of the olfactory epithelium, amplify centrioles from an initial pair. The progenitor cell divides and the daughter cell differentiates into an immature OSN, which extends a dendrite towards the apical surface. Centrioles migrate in multiple groups within a single dendrite in immature OSNs. The leading group of centrioles moves in tandem with dendrite extension and can be followed by lagging groups. Centriole groups move relatively slowly towards the apical end of the dendrite. The maximum rate observed indicates that centrioles could take greater than 9.26 hr to traverse an epithelium 100 µm thick. This timing is well within the multiple days required for dividing progenitors to give rise to fully mature OSNs (*Rodriguez-Gil et al., 2015*; *McClintock et al., 2020*). During migration, most of the centrioles retain markers associated with immature centrioles and lack the appendages required to make cilia. The exception is a single migrating centriole, presumably an original parental centriole, which has

appendages and is able to form a single cilium in the dendritic knob, prior to the others. Finally, most centrioles acquire distal and subdistal appendages and form the dozens of olfactory cilia found in the mature OSN. The parental centriole retains structural characteristics that differentiate it from the other centrioles within the same cell.

We previously showed that centrioles in OSNs are amplified, at least in part, by formation of centriole rosettes in progenitor cells, without deuterosomes like those seen in MMCCs (*Ching and Stearns, 2020*). Here, we exploited the resolution afforded by expansion microscopy to precisely count centrioles, finding that the number of centrioles during migration and in mature OSNs was even higher than previously reported, with a mean of approximately 36 centrioles in the latter. The number of centrioles was similar for both states, suggesting that all centriole amplification takes place prior to the initiation of migration and dendrite extension. This is a higher number of centrioles than reported by *Uytingco et al., 2019* and *Ukhanov et al., 2022*; this could be due to differences in microscopy and counting methods, regional differences in the mouse olfactory epithelium (*Challis et al., 2015*), or variation between animals. Based on the new evidence we present here, a parsimonious explanation for how OSNs achieve their final centriole number is that amplification is initiated by rosette formation around the two parental centrioles in progenitor cells, followed by a single mitotic division and centriole disengagement from the rosettes, and further amplification in post-mitotic cells that become OSNs.

OSNs are, to our knowledge, unique in that they undergo a mitotic division with amplified centrioles, which is then followed by further amplification (*Ching and Stearns, 2020*). Thus, an immature OSN potentially has centrioles of three different ages: the parental centriole, amplified centrioles formed prior to mitosis, and those that may have formed after mitosis. We found that some of the amplified centrioles in immature OSNs have nucleation-associated proteins of the PCM, including gamma-tubulin, CDK5RAP2, and pericentrin, consistent with a portion of the amplified centrioles having undergone the centriole-to-centrosome conversion as described in cultured mammalian cells (*Wang et al., 2011*; *Izquierdo et al., 2014*). Similarly, some of the centrioles in immature OSNs have the cartwheel components SASS6 and STIL, as might be expected for those centrioles formed after the mitotic division, whereas all centrioles lack these components in mature OSNs.

Centrioles in mature OSNs have the proteins associated with rootlets, distal appendages, and subdistal appendages/basal feet, typical of centrioles underlying cilia in other multiciliated cells (*Anderson and Brenner, 1971*; *Menco et al., 1978*; *McClintock et al., 2008*; *Sorokin, 1968*). It is interesting that the final modification of the non-parental centrioles with these components in OSNs appears to be controlled spatially and/or temporally, by their arrival in the forming dendritic knob, as they were absent on migrating centrioles in immature OSNs. This might occur, for example, by localized translation or accumulation of the protein components of these structures, or by temporal control of the expression of those components. In cycling mammalian cells grown in culture, centriole maturation is driven by cell cycle progression and the activity of PLK1 kinase (*Kong et al., 2014*; *Shukla et al., 2015*). In MMCCs, low-level activation of mitotic kinases is important for centriole amplification (*Al Jord et al., 2017*). In OSNs, it is possible that centriole maturation is also controlled by PLK1 and other mitotic regulators, with the added element of spatiotemporal control. Interestingly, it has also been shown that loss of centrin-2 results in fewer cilia on OSNs as well as more OSNs with centrioles below the apical surface (*Ying et al., 2014*). This would be consistent with a defect in acquisition of appendages, although the mechanism by which centrin-2 affects centriole docking is unclear.

The presumptive parental centriole differed from all others in OSNs in having subdistal and distal appendage proteins prior to its migration to the dendrite tip. This was apparent during migration of centrioles and in the early stage of cilium formation when only this centriole was associated with a cilium. The presence of a single cilium before formation of the full complement of multiple cilia might enable signaling important for OSN differentiation, as has been proposed for the primary cilium of MMCCs that precedes multiciliated differentiation (*Jain et al., 2010*), although we note that *Schwarzenbacher et al., 2005* found that the earliest OSN cilia formed lack olfactory receptors. The parental centriole retained a radial array of the subdistal appendage component centriolin even in the mature OSN, whereas all other centrioles had a single focus of centriolin, consistent with them having basal feet. This is remarkably similar to the situation in MMCCs in which a single parental centriole also differed from other centrioles and was associated with what was termed a 'hybrid cilium' (*Liu et al., 2020*). It is unclear what prevents the parental centriole from forming a cilium prior to reaching the

dendritic knob, although it might be due to the same spatiotemporal regulation that we postulated to control centriole maturation.

How do centrioles migrate from the cell body to the tip of the dendrite? Our work rules out several possibilities. First, centrioles within a single cell often travel in multiple groups separated by relatively long distances along the dendrite. This excludes the possibility that forces act upon only a single centriole, for instance, the parental centriole. Second, our data also exclude the possibility that centrioles might migrate by docking to membranes as only the presumptive parental centriole bears the distal appendages required for membrane association during migration. Third, migrating centrioles lack rootletin, ruling out models in which centrioles are moved by attachment of centriolar rootlets to actin filaments, as observed during macrociliary cell development in the comb jelly *Beroe* (*Tamm and Tamm, 1988*).

Our data support a model in which the microtubule cytoskeleton is critical for centriole migration. Using a novel explant system, we found that inhibiting microtubule dynamics disrupts centriole migration to the apical surface. In this respect, OSNs are similar to MMCCs of the oviduct, in which paclitaxel-mediated stabilization of microtubules does not affect centriole biogenesis, but can inhibit centriole migration (*Lemullois et al., 1988*; *Boisvieux-Ulrich et al., 1989*). Because of the very different cell and tissue architectures in MMCCs compared to OSNs, it is unclear whether the mechanism of inhibition by microtubule stabilization is the same in these cell types. Our establishment of an explant system will be useful for future work in distinguishing the exact mechanisms by which cytoskeletal elements influence centriole migration in OSNs.

How do forces act upon migrating centrioles? One possibility is that microtubule motors are directly involved in centriole transport. In the *C. elegans* ciliated PQR neuron, SAS-5, an ortholog of STIL, directly interacts with a dynein light chain (DLC-1/LC8), suggesting that centrioles may be cargos of cytoplasmic dynein in the dendrite (*Li et al., 2017*). In addition, the rate of migration for the single centriole that migrates in *C. elegans* neurons is similar to that of mouse OSN centrioles (in OSNs: maximum of 0.18 µm/min; in *C. elegans*: 0.076 µm/min, *Li et al., 2017*), suggesting that similar mechanisms may drive centriole migration in both contexts. However, dynein motors acting processively exhibit velocities much higher than those observed in OSNs or the PQR neuron, approximately 60 µm/min in cells and *in vitro* (*Presley et al., 1997*; *King and Schroer, 2000*). Thus, dynein might act non-processively and/or interact with centrioles transiently in OSNs. This is reminiscent of slow axonal transport, which drives the anterograde transport of cytoskeletal elements within neuronal axons (*Baas and Buster, 2004*; *Roy, 2020*). Although mechanisms of fast and slow axonal transport were once thought to be distinct, more recent work has indicated that cargos undergoing slow axonal transport interact with the same type of molecular motors, though intermittently (*Brown, 2014*; *Roy, 2020*). Prolonged pauses in transport of centrioles due to motor disengagement or reaching the end of a microtubule track could cause 'traffic jams' in the dendrite, resulting in centrioles appearing as groups. Hence, it is plausible that transient interactions similar to slow axonal transport are responsible for centriole movement in OSN dendrites.

Our finding that microtubule nucleation proteins are present in migrating centrioles suggests another, non-mutually exclusive hypothesis: that migrating centrioles are microtubule organizing centers, with self-nucleated microtubules involved in their motility. Force could be derived from microtubule polymerization itself or by interaction of nucleated microtubules with an existing dendritic microtubule array. In other mammalian neurons, centrioles remain near the nucleus and lose microtubule nucleating proteins as the cell differentiates (*Leask et al., 1997*; *Stiess et al., 2010*; *Yonezawa et al., 2015*; *Sánchez-Huertas et al., 2016*; *Wilkes and Moore, 2020*). In cycling mammalian cells, PCM nucleates microtubule asters that center and position the centrosome. This occurs through pushing forces driven by microtubule nucleation, and pushing and pulling forces driven by molecular motors (reviewed in *Burakov and Nadezhdina, 2020*). Similarly, microtubule nucleation may help orient and position centrioles toward the apical surface in OSNs. This would be consistent with a model in which forces are acting upon multiple centrioles, perhaps even individual centrioles, within the same dendrite.

In addition to migration within dendrites, the PCM of OSN centrioles might also influence dendrite microtubule organization itself. OSN dendrites have a polarized microtubule cytoskeleton, in which the microtubule minus ends are oriented out, toward the dendritic tips (*Burton, 1985*). This polarity, observed in bullfrog mature OSNs, is in contrast to dendrites of many neurons, which have a mixed

polarity. Although the microtubule polarity is not known in the immature mouse OSNs that we study here, an attractive possibility is that the centrioles positioned at the tip of the dendrite function as a centrosome, nucleating microtubules, with their minus ends out. Supporting this hypothesis, we observed that PCM proteins are present at centrioles in both immature and mature OSNs. An interesting parallel is that microtubule polarity in dendrites of non-ciliated *C. elegans* PVD neurons is established by a non-centrosomal microtubule organizing center that is present at the dendritic tip (*Liang et al., 2020*), suggesting that this may be a conserved mechanism of establishing and maintaining microtubule polarity in dendrites.

Finally, we have focused here on microtubule-based mechanisms, but acknowledge that other mechanisms, for example, based on the actin cytoskeleton, might also be important for centriole migration and docking in OSN dendrites. We expect that the new techniques for manipulating and studying the olfactory epithelium that we describe will facilitate further research into these and other questions about how multiple centrioles and cilia are formed in OSNs and participate in their structure and function.

## Materials and methods

### Mice

Olfactory epithelium samples were taken from both male and female mice, euthanized by $CO_2$ inhalation, for all experiments. Mice were between 1 and 6 months old. (See each section for specific ages.) Mice constitutively overexpress eGFP-centrin2 to mark centrioles, and Arl13b-mCherry to mark primary cilia (line generated by *Bangs et al., 2015*). In no instance did we observe Arl13b-mCherry in olfactory cilia, but Arl13b-mCherry was present in primary cilia of olfactory epithelium progenitor cells. Native fluorescence signal from eGFP and mCherry was not preserved in expansion microscopy.

Cell lines hTERT RPE-1 cells were cultured in DMEM/F-12 (Corning) supplemented with 10% Cosmic Calf Serum (CCS; HyClone). All cells were cultured at 37°C under 5% $CO_2$ and are mycoplasma-free.

### Microscopy

All confocal microscopy images were acquired as single planes or *Z*-stacks collected at 0.27 μm intervals using a Zeiss Axio Observer microscope (Carl Zeiss) with a PlanApoChromat 63×/1.4 NA objective,

**Table 1.** Antibodies used in this study.

| Target | Source | Dilution |
|---|---|---|
| Acetylated tubulin | Sigma-Aldrich, clone 6-11B-1, RRID:AB_477585 | 1:1000 |
| CDK5RAP2 | Millipore, 06-1398 rabbit polyclonal, RRID:AB_11203651 | 1:200 |
| Centrin | EMD Millipore, clone 20H5, RRID:AB_10563501 | 1:1000 |
| CEP164 | Rabbit polyclonal previously described in *Lee et al., 2014* | 1:500 |
| GFP | Rabbit antibody previously described in *Hatch et al., 2010* | 1:2000 |
| GAP43 | Novus Biologicals, NB300, rabbit polyclonal, RRID:AB_921392 | 1:250 (Blocking buffer: PBSBT. 2% BSA, 0.1% Triton X-100, 1x PBS) |
| Pericentrin | BD Biosciences, clone 30, RRID:AB_399294 | 1:500 |
| Centriolin | Santa Cruz Biotech, clone C-9, RRID: AB_10851483 | 1:50 |
| STIL | Abcam, rabbit polyclonal, RRID:AB_2197878 | 1:1,000 |
| SASS6 | Santa Cruz Biotech, RRID:AB_1128357 | 1:200 |
| ODF2 | Novus Biologicals, mouse IgG2a, RRID:AB_1146453 | 1:200 |
| Rootletin (CROCC) | Santa Cruz Biotech, clone C-2, RRID:AB_10918081 | 1:100–200 |
| β-Tubulin III | BioLegend, clone TuJ1, RRID:AB_2313773 | 1:1000–2000 |
| Gamma-tubulin | Sigma-Aldrich, clone GTU88, RRID:AB_532292 | 1:1000 |

a Yokogawa CSU-W1 head, and a Photometrics Prime BSI express CMOS camera. Slidebook software (Intelligent Imaging Innovations, 3i) was used to control the microscope system.

## Antibodies

Primary antibodies used for immunofluorescent staining are listed in *Table 1*. Alexa Fluor-conjugated secondary antibodies (Thermo Fisher) were diluted 1:1000 for standard immunofluorescence and 1:500 or 1:1000 for expansion microscopy.

## Immunofluorescent staining of cryosections

Olfactory epithelia were dissected mechanically from mice that were 1–4 months old. All stainings were performed at least twice, in olfactory epithelium taken from both a male and a female mouse. After initial snout removal, fine dissection was performed in a dish of cold Tyrode's solution (140 mM NaCl, 5 mM KCl, 10 mM HEPES, 1 mM CaCl$_2$, 1 mM MgCl$_2$, 1 mM sodium pyruvate, 10 mM glucose in ddH$_2$O), as in other reports (*Oberland and Neuhaus, 2014*) and at http://github.com/katieching/Protocols. Whole olfactory epithelia, turbinate epithelia, or septum epithelia were fixed immediately in 4% PFA in phosphate buffered saline (PBS) at 4°C for 3–24 hr. Samples were then washed in PBS and stored at 4°C. Before mounting, samples were equilibrated in 1–5 mL of 30% sucrose solution in water for a minimum of 12 hr at 4°C. Samples were embedded in OCT compound (Sakura Tissue-Tek) on dry ice and stored at –80°C. Embedded samples were sectioned at 14 µm on a Leica cryostat and adhered to charged slides by drying at room temperature for approximately 1 hr. Slides were stored with drying pearls (Thermo Fisher) at –80°C and thawed under desiccation no more than twice. Sections were outlined with a hydrophobic pen. Residual-free aldehydes were quenched while samples were rehydrated in PBS with 0.3 M glycine, 1% calf serum, and 0.1% Triton-x 100 for 0.5 hr at room temperature. Samples were blocked and permeabilized for an additional 0.5–4 hr in PBS with 1% calf serum and 0.1% Triton-x 100. Antibodies were also diluted in PBS with 1% calf serum and 0.1% Triton-x 100 (see *Table 1*). Slides were incubated with primary antibodies for approximately 3 hr at room temperature or overnight at 4°C, washed in PBS, incubated with secondary antibodies for approximately 1 hr at room temperature, washed in PBS, incubated in DAPI diluted 1:1000 for 5 min, washed in PBS, and mounted in MOWIOL. Where phalloidin conjugated to Alexa Fluor 568 was used, it was included at the secondary antibody step at a 1:1000 dilution (Molecular Probes). Where EdU was used, incorporation was visualized using the Click-iT Alexa Fluor 594 kit according to the manufacturer's instructions (Molecular Probes). Slides were imaged by spinning disk confocal microscopy, and images were acquired with the SlideBook software by Intelligent Imaging Innovations (3i). Images were processed in Fiji (*Schindelin et al., 2012*). Images are rotated to orient the apical surface toward the top of the page. For images with high background, contrast in the representative images was adjusted uniformly across the image such that the area outside of cells was black and areas of high signal were just below saturation. Dendrites stained for GAP43 were measured in ImageJ (*Schindelin et al., 2012*; details at http://github.com/katieching/Protocols), and lengths were plotted using Statistika (*Figure 2B*; *Weissgerber et al., 2017*).

## Expansion microscopy – olfactory epithelium

Details about expansion microscopy for olfactory epithelium can be found at http://github.com/katieching/Protocols and were based on work by *Gambarotto et al., 2019* and *Sahabandu et al., 2019*. In brief, samples from 1- to 4-month-old mice were dissected mechanically in a dish of cold Tyrode's solution (140 mM NaCl, 5 mM KCl, 10 mM HEPES, 1 mM CaCl$_2$, 1 mM MgCl$_2$, 1 mM sodium pyruvate, 10 mM glucose in ddH$_2$O), fixed, washed, and cryoprotected by the same methods used for unexpanded immunofluorescent staining. All stainings were performed at least twice in olfactory epithelium taken from both a male and a female mouse. Samples were embedded in OCT (Sakura Tissue-Tek) on dry ice and stored at –80°C. Embedded samples were sectioned at 14 µm thickness on a Leica cryostat, placed on charged glass slides within a border pre-drawn with hydrophobic pen, and allowed to thaw for 1 min. Sections were then washed three times in PBS for approximately 5 min per wash. PBS was removed, and sections were incubated in a monomer fixative solution (0.7% formaldehyde and 1% w/v acrylamide in water) for 2 hr at room temperature. Sections were washed once in monomer solution (19% w/v sodium acrylate, 10% w/v acrylamide, and 0.1% BIS in PBS). Solution was removed, and slides containing sections were inverted onto droplets of cold gelation

solution (monomer solution with TEMED and ammonium persulfate added to a final concentration of 0.5% each). Stacks of approximately five coverslips were used as spacers to achieve the desired gel thickness. Gels were set for 5 min on ice and then overnight at room temperature in a sealed chamber with wet paper towels. Slides with gels attached were incubated at 95°C in a large volume of denaturation buffer (200 mM SDS, 200 mM NaCl, and 50 mM Tris in water) for approximately 4–6 hr. Gels were washed and then pre-expanded in water overnight. Gels were then incubated in PBS for at least 30 min. Gels were incubated in primary antibody solution in PBS with or without 3% bovine serum albumin and 0.1% Triton-x100 (see *Table 1*) overnight on a nutator at 4°C. Gels were washed three times in PBS for 10–30 min per wash and then incubated in a solution of secondary antibodies conjugated to Alexa Fluors plus DAPI, diluted 1:500 or 1:1000, overnight with gentle agitation at 4°C. Gels were washed in PBS for at least 30 min, then in water three times for 30 min per wash. Gels were then allowed to fully expand in water for at least an hour before mounting in a glass-bottom imaging dish and imaging by spinning disk confocal microscopy. Native fluorescence signal from eGFP and mCherry was not preserved. As needed, gels were imaged in a poly-lysine-coated dish to reduce sample movement during acquisition. Centriole counts were semi-automated: an initial volume segmentation was performed in Imaris, followed by manual inspection. Graphs for *Figure 1—figure supplement 1* were made using GraphPad Prism.

## Expansion microscopy – tissue culture cell lines

hTERT RPE-1 cells were grown to confluency on 12 mm coverslips. Cells were fixed with methanol at –20°C for 10 min, then washed with 1× PBS. PBS was removed and cells were incubated in monomer fixative solution (0.7% formaldehyde and 1% w/v acrylamide in water) overnight at 37°C. Coverslips were then inverted onto droplets of cold gelation solution (19% w/v sodium acrylate, 10% w/v acrylamide, 0.1% BIS, 0.5% TEMED, 0.5% ammonium persulfate in PBS) and allowed to set for 5 min on ice, then at 37°C for 1 hr. Gels were incubated in denaturation buffer (200 mM SDS, 200 mM NaCl, and 50 mM Tris in water) for 45 min at 95°C, then washed in water and expanded in water overnight. Gels were then incubated in PBS for at least 30 min. Gels were incubated in primary antibody solution in PBS with or without 3% bovine serum albumin and 0.1% Triton-X100 (see *Table 1*) overnight on a nutator at 4°C. Gels were washed three times in PBS for 10–30 min per wash and then incubated in a solution of secondary antibodies conjugated to Alexa Fluors plus DAPI, diluted 1:500 or 1:1000, overnight with gentle agitation at 4°C. Gels were washed in PBS for at least 30 min, then in water three times for 30 min per wash. Gels were then allowed to fully expand in water for at least an hour before mounting in a glass-bottom imaging dish and imaging by spinning disk confocal microscopy.

## Live imaging of olfactory epithelium

Septal or flattened turbinate olfactory epithelium was mechanically dissected from 3.5- to 6-month-old mice in the same manner as for explant preparation. Imaging was performed in samples taken from three different animals and included both male and female mice. The procedure for mounting was partly based on work from *Williams et al., 2014*. For this study, dissected samples were pulled onto a coverslip containing an agarose pad made of 1% low-melt agarose (Lonza #50100) dissolved in Tyrode's buffer. The coverslip with the sample was then inverted into the center of an imaging dish containing 150 µL of imaging buffer (Tyrode's with 5% cosmic calf serum and 5 µg/mL Hoechst 34580 [Invitrogen #H21486], mixed by pipette and vortexed) and allowed to sit for at least 5 min before imaging. Coverslips were sealed into place with melted Vaseline as needed. Samples were imaged *en face* by spinning disk confocal microscopy at approximately 37°C with low laser power until no more than 4 hr after euthanasia. Files were acquired and scored using the SlideBook software by Intelligent Imaging Innovations (3i). For scoring, centriole groups below the apical surface were identified in a 4D volume view and measured in a maximum intensity projection over the y-axis with respect to a group of nearby mature OSN centrioles as a fiducial mark. Locations for each group were scored in the first and last frame during which they were visible in the maximum projection, and average rates were calculated from the difference between those measurements. Kymograph analysis was performed using Dynamic Kymograph in Fiji (*Zhou et al., 2020*).

## Culture and analysis of septum explants grown on air-liquid interface

Details about growing and scoring olfactory epithelium explants can be found at http://github.com/katieching/Protocols and are partly based on the procedure described by *Oberland and Neuhaus, 2014*. In brief, septum epithelia from 1- to 4-month-old mice were dissected mechanically in cold Tyrode's buffer and placed apical side up on transwell polycarbonate filters with 0.4 µm pore size with the bottom compartment containing additional Tyrode's buffer. All experiments were performed in explants taken from both male and female mice. Once all samples were placed, buffer was aspirated from the bottom compartment and replaced with warm explant medium (15% fetal bovine serum and 0.02% L-ascorbic acid in Waymouth's media; Thermo Fisher Scientific #11220035; based on *Farbman, 1977*). No medium was added to the top compartment, preserving the air-liquid interface. Relevant treatments (10 µM EdU [Thermo Fisher Scientific #A10044], 15 µM paclitaxel [EMD Millipore #580555], 10 µg/mL nocodazole [Thermo Fisher #AC358240500], or an equivalent volume of DMSO) were mixed into media by pipetting and vortexing before addition. Samples were incubated in a 33°C, 5% $CO_2$, humidified incubator for 1 hr or 6 hr before forceps were used to transfer them from the filters to 4% PFA in PBS for overnight fixation. For *Figure 5B and C*, samples within each replicate were obtained from the same animal. For *Figure 5D–F*, samples were matched such that explants taken from the two conditions came from the same animal at the same time point. Where explants were taken from the same animal, conditions were randomized between the right and left halves of the septum. Samples were then washed and processed by the same method of immunofluorescent staining described above. To score explants (*Figure 5F*, *Figure 5—figure supplement 1B and C*), sample size was selected by (1) measuring the average length between areas of high amplification (approximately 250 µm), (2) scoring a length greater than that for the condition with the most limited sample, and (3) scoring progressively (i.e., not selecting regions) along the epithelium of all other conditions until the length was matched. For all experiments except nocodazole treatment, this was then repeated in samples from the opposite sex. Differences in the number of centriole groups were evaluated by a paired permutation test in R in which each category (e.g., animal 1, number of subapical groups at t = 1 hr) was treated as a pair of data points (drug vs. control), and resampling was simulated 10,000 times. Code can be found at https://github.com/katieching/Protocols, (copy archived at swh:1:rev:89f322e1fdcbe56f7d569fd670715f7dfa436eeb; *Ching, 2022*).

## Acknowledgements

We thank members of the Stearns lab for helpful feedback and suggestions, and Moe Mahjoub for information about the centriolin antibody. This project was supported by the National Science Foundation Graduate Research Fellowship under Grant No. D-G16E5 6518 (to KC) and by the NIGMS of the National Institutes of Health under award numbers R35 GM130286 and R01 NS082208 (to TS), K99 GM131024 (to JTW), and T32GM007276 (CMB Training Grant).

## Additional information

### Funding

| Funder | Grant reference number | Author |
| --- | --- | --- |
| National Institutes of Health | R35GM130286 | Tim Stearns |
| National Institutes of Health | R01NS082208 | Tim Stearns |
| National Institutes of Health | K99GM131024 | Jennifer T Wang |
| National Institutes of Health | T32GM007276 | Kaitlin Ching |
| National Science Foundation | D-G16E5 6518 | Kaitlin Ching |

| Funder | Grant reference number | Author |
|---|---|---|

The funders had no role in study design, data collection and interpretation, or the decision to submit the work for publication.

## Author contributions
Kaitlin Ching, Jennifer T Wang, Conceptualization, Investigation, Writing – original draft, Writing – review and editing; Tim Stearns, Conceptualization, Funding acquisition, Supervision, Writing – original draft, Writing – review and editing

## Author ORCIDs
Kaitlin Ching http://orcid.org/0000-0002-0517-2421
Jennifer T Wang http://orcid.org/0000-0002-8506-5182
Tim Stearns http://orcid.org/0000-0002-0671-6582

## Ethics
All animal procedures in this study were approved by the Stanford University Administrative Panel for Laboratory Animal Care (SUAPLAC protocol 11659) and carried out according to SUAPLAC guidelines.

## Decision letter and Author response
Decision letter https://doi.org/10.7554/eLife.74399.sa1
Author response https://doi.org/10.7554/eLife.74399.sa2

---

# Additional files

## Supplementary files
• Transparent reporting form

## Data availability
Figure 2 - Source Data 1, Figure 5 - Source Data 1, Figure 5 - Figure supplement 1 - Source Data 1, and Figure 5 - Figure supplement 1 - Source Data 2 contain the numerical data used to generate the figures.

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
