## [Editor Report]

This work will interest cell and developmental biologists studying centriole biogenesis, cilia assembly, neuron biology and development of the olfactory system. Using expansion microscopy as well as new culturing and live imaging techniques, the authors provide an unprecedented view of centriole duplication, migration, and maturation in mouse adult olfactory sensory neurons (OSNs). The major advance here is mostly technical, but the study beautifully lays the groundwork for using OSNs as a model to understand the cell biological underpinnings of organelle assembly, motility and maturation in an adult mouse tissue *in vivo*.

---

## [Decision Letter]

**Decision letter after peer review:**

Thank you for submitting your article "Long-range migration of centrioles to the apical surface of the olfactory epithelium" for consideration by *eLife*. Your article has been reviewed by 3 peer reviewers, one of whom is a member of our Board of Reviewing Editors, and the evaluation has been overseen by Piali Sengupta as the Senior Editor. The reviewers have opted to remain anonymous.

Essential revisions:

All three reviewers were impressed by the technical advances and beautiful data obtained by expansion microscopy, but felt the study was primarily descriptive and needed some additional mechanistic insight for publication in *eLife*. In consultation, the reviewers felt it would be most straightforward to better define the role of microtubules in centriole migration. Specifically,

1) Extend inhibitor treatments to include nocodazole.

2) Define the orientation of MT plus and minus ends during migration. This would indicate whether migrating centrioles nucleate MTs in an oriented manner and test the hypothesis that "migrating centrioles are microtubule organizing centers, with self-nucleated microtubules involved in their motility".

3) Provide a more in depth characterization of the relationship between dendrite extension and centriole migration, as the reviewers felt these data were less convincing than data throughout the rest of the paper.

Full reviews from all three reviewers are included below.

*Reviewer #1 (Recommendations for the authors):*

1. The live imaging experiments measure the centriole migration rate as 0.18 μm per min. So it should take about 9 hours for them to traverse the whole distance. Is that timing consistent with the known development of these neurons and other observations? Can the authors comment on this in the discussion?

2. Figure 2B examines centrioles in immature OSNs to determine whether migration coincides with, or follows dendrite extension. I'm not really sure what I'm looking at in these images. It is not very clear that this is an extending/not fully extended dendrite. Compared to other images in this paper, this one is not particularly convincing.

3. Figure 5 characterizes the migration defect upon paclitaxel treatment, but I feel this data could be improved in its presentation. Provide an indication of where the boundaries of basal, middle and subapical positions are scored within a schematic or better yet representative image. Also providing the number of centrioles in each category at time=0 would also be helpful. Is the a decrease in the number of fully apically-docked centrioles? I'm trying to get an idea of what percentage of centrioles would have been migrating during this 6 hour time period and how many of these are affected by the MT stabilizing drug.

4. Is the mature, presumptive parental, centriole ever observed in a lagging cluster? Or is there reason to think it initiates the migration?

5. To address concerns about the descriptive nature of the study, and provide some mechanistic insight into some aspect of this fascinating cell biology, the discussion mentions the possibility that Plk1 could control the timing of centriole maturation. Inhibitors of cell cycle kinases work well in explant cultures. The authors could test this hypothesis directly in explant culture, and I don't think it would be beyond the scope of the current study.

*Reviewer #2 (Recommendations for the authors):*

1. The ALI culture is really impressive and important but I think needs a little more data to justify its usefulness. While the authors have shown nicely that there is still some proliferation (based on EdU), I am not sure that they have sufficiently shown that the whole migration process is actually happening. While this is somewhat inferred from the taxol data it is an important aspect to this experimental paradigm that should be more clearly shown.

2. The MT analysis seems very preliminary. "Our finding that microtubule nucleation proteins are present on migrating centrioles suggests another, non-mutually exclusive hypothesis: that migrating centrioles are microtubule organizing centers, with self-nucleated microtubules involved in their motility"….I think this is very likely and really should be addressed. I think a deeper analysis (see point 3) and additional drug treatments (eg nocodazole) are necessary. While genetic analysis of motors would be exciting and likely to yield interesting results it is appreciated that this is outside the scope of this manuscript.

3. OSN dendrites have a polarized microtubule cytoskeleton, in which the microtubule minus ends are oriented out, toward the dendritic tips (Burton, 1985)…While Burtons work is certainly beautiful, I think the relevant question here is what is the orientation of MTs during migration. One might not expect (-) ends at the OSN dendrite until centrioles arrive. Do migrating centrioles nucleate MTs in an oriented manner? This seems critical to determine the role of MTs in centriole migration. Additionally, what does the MT network look like during this centriole migration. I think better imaging of MTs/(-) end markers and (+) end markers would help explain the role of MTs in this process better.

4. Do the dendritic knobs position properly in the absence of centrioles / MTs. Perhaps this is known but it was unclear to me from the data and it seems important to know if the dendrites are positioned prior to centriole migration or do the first wave of centrioles act as leaders that push (or pull) the dendrite to its proper site (as is sort of drawn in the cartoon).

5. "the subapical region, and the middle and basal region (regions marked in Figure 1A)" ….please explain how these were defined experimentally.

6. The quantifications need statistical analysis.

*Reviewer #3 (Recommendations for the authors):*

One of the most interesting aspects highlighted by the authors in this and their previous study is the atypical regulation of centriole assembly in olfactory neuron progenitors. However, some important aspects of these mechanisms remain unclear. As proposed by the authors in the discussion paragraph, it is possible that centrioles are also assembled in immature OSNs. The migrating centrioles would then belong to 3 different generations – a mother centriole, daughter centrioles formed in the previous cycle, and procentrioles formed in the OSNs. This could explain that the number of centrioles is slightly lower in immature OSNs than in mature ones (although this might result from experimental issues, as in Supplementary Figure 1A, centriole labeling appears sometimes incomplete and difficult to interpret), that only some of the migrating centrioles still possess SAS6 and STIL, and that migrating centrioles appear to vary in length (Supplementary Figure 1A, Figure 2A, 3B).

It would be important to establish whether STIL and SAS-6 labeled centrioles can be formed at the previous cell cycle, or whether they are formed in immature OSNs. Is it possible to analyze immature OSNs in G1 phase? Alternatively, could they use centrinone to block centriole duplication in immature OSNs and determine if this leads to a loss of STIL/SAS6 signaling during migration?

– The authors propose that there may be spatial control of centriole maturation (e.g., line 393). However, it is also possible that this control is simply temporal. In the first case, inhibition of dendrite extension should block centriole maturation. Is it possible to block dendrite extension and/or centriole migration, for example by using drugs that depolymerize actin or microtubules?

Regarding the mechanisms underlying centriole migration, the observation that dendrite extend concomitantly to centriole migration is interesting but could be more substantiated. Ideally, this would involve specifically blocking one or the other, which is probably difficult to achieve in this context. Again, is it possible to use cytoskeleton drugs against actin or microtubules to do so? Alternatively, do leading centriole groups sometimes move backwards, and if so, does the dendrite continue extending?

– The absence of rootletin labeling in centrioles is used as an argument to exclude a rootlet involvement in migration. However, the labeling is also rather weak in centrioles of mature OSNs. Is it known that the rootlet is poorly developed in these cells (e.g., as seen by EM), or is this due to labeling issues with the antibody used in this study? In this case, it is not clear that developing rootlets would be detected efficiently in migrating centrioles.

---

## [Author Response]

Essential revisions:All three reviewers were impressed by the technical advances and beautiful data obtained by expansion microscopy, but felt the study was primarily descriptive and needed some additional mechanistic insight for publication in eLife. In consultation, the reviewers felt it would be most straightforward to better define the role of microtubules in centriole migration. Specifically,1) Extend inhibitor treatments to include nocodazole.

We agree with the reviewers that it would be interesting to define the role of the microtubule cytoskeleton in centriole migration. We have tried nocodazole treatment in our explant system, and found that 6 hours of 10 μg/mL nocodazole had a minor effect on centriole migration. These results are summarized in a new supplementary figure (Figure 5 – Supplementary Figure 1). However, it is important to note that in neurons, nocodazole treatment is insufficient to depolymerize all microtubules due to the presence of stabilizing factors (Baas et al., Cytoskeleton 2016). It is likely that these olfactory sensory neurons also harbor a stable population of microtubules that are resistant to nocodazole treatment. We conclude that further work is needed to define the relative roles of the stable and labile microtubule populations in centriole migration.

2) Define the orientation of MT plus and minus ends during migration. This would indicate whether migrating centrioles nucleate MTs in an oriented manner and test the hypothesis that "migrating centrioles are microtubule organizing centers, with self-nucleated microtubules involved in their motility".

We thank the reviewers for highlighting this important question. The definitive experiments to address microtubule orientation in OSN dendrites are outside the scope of the current manuscript, as they would involve challenging experiments either to express a fluorescently-tagged microtubule end-binding protein to observe microtubule growth in live OSNs, or to use microtubule hooking experiments in which microtubule polarity in immature OSNs might be observed by electron microscopy. However, we have performed immunofluorescence experiments on fixed tissues to address this question to the best of our ability. In the original manuscript, we find that γ-tubulin is present at migrating centrioles, which marks the location of some microtubule minus ends. In attempting to define the location of microtubule plus-ends, we found that EB1/Mapre1 is expressed in iOSNs using a previously published scRNAseq dataset (Fletcher et al., Cell Stem Cell, 2017). However, a widely-used EB1 antibody did not result in any specific staining in the olfactory epithelium, despite its ability to detect microtubule plus-ends in NIH3T3 cells grown in culture. We summarize these results in Author response image 1.

**Author response image 1. sa2fig1:** EB1 staining in cultured NIH3T3 cells and olfactory epithelium. (**A**) Left: Mitotic 3T3 cell. Right: Interphase 3T3 cell. EB1 (green) and DAPI (blue). Both images are maximum z-projections of confocal stacks. (**B**) Olfactory epithelium. EB1 (white), β- tubulin III (magenta), eGFP-centrin2 (green), and DAPI (blue). The dendrite is outlined in white. Images are maximum projections of confocal stacks.

3) Provide a more in depth characterization of the relationship between dendrite extension and centriole migration, as the reviewers felt these data were less convincing than data throughout the rest of the paper.

We agree that it is important to improve the imaging of dendrites in the olfactory epithelium and better characterize the relationship between dendrites and migrating centrioles. We were able to obtain another GAP43 antibody that greatly improves the visualization of OSN dendrites (antibody previously used in Pluznick et al., PLoS One, 2011 and Rodriguez-Gil et al., J Comp Neurol, 2008). The results are included in a new version of Figure 2. In brief, we find that migrating centrioles lag behind the growth cone by as much as 8.4 μm (average=3.68 μm). We have edited the text in accordance with these new results.

Full reviews from all three reviewers are included below.Reviewer #1 (Recommendations for the authors):1. The live imaging experiments measure the centriole migration rate as 0.18 μm per min. So it should take about 9 hours for them to traverse the whole distance. Is that timing consistent with the known development of these neurons and other observations? Can the authors comment on this in the discussion?

We thank the reviewer for this comment. We believe that this timing is consistent with the known development of these neurons, (McClintock et al., Chemical Senses 2020, Rodriguez-Gil et al., PNAS 2015). We have included text about this in the discussion.

2. Figure 2B examines centrioles in immature OSNs to determine whether migration coincides with, or follows dendrite extension. I'm not really sure what I'm looking at in these images. It is not very clear that this is an extending/not fully extended dendrite. Compared to other images in this paper, this one is not particularly convincing.

We agree that the original Figure 2B was not convincing, and have now included a new figure with a new GAP43 antibody. We have also edited the text, as explained above for essential revision 3.

3. Figure 5 characterizes the migration defect upon paclitaxel treatment, but I feel this data could be improved in its presentation. Provide an indication of where the boundaries of basal, middle and subapical positions are scored within a schematic or better yet representative image. Also providing the number of centrioles in each category at time=0 would also be helpful. Is the a decrease in the number of fully apically-docked centrioles? I'm trying to get an idea of what percentage of centrioles would have been migrating during this 6 hour time period and how many of these are affected by the MT stabilizing drug.

We had mistakenly neglected to include this information in the original manuscript, and have now included a new figure, Figure 5 – Supplementary Figure 1A, that indicates the boundaries between the middle/basal and the subapical groups. We believe the best comparison is between t=1 h vs t=6 h, because the t=1 h time point represents a reasonable time for the explant to equilibrate to the culture conditions. Note that these are paired samples, so it was not possible to have a t=0 time point, however, we have also included a new figure, Figure 5 – Supplementary Figure 1B that shows t=0 samples from different mice. Also, we note that the question as to whether fewer centrioles are fully docked under inhibitor treatment conditions is interesting and important, but we have not been able to address that here due to (1) the technical challenge of imaging the apical surface (requires phalloidin staining) and centriole docking (requires expansion microscopy, or preferably an even higher resolution method) simultaneously, and (2) the time points used, which are likely too short to capture loss of mature OSNs and the resulting population-level changes at the apical surface.

4. Is the mature, presumptive parental, centriole ever observed in a lagging cluster? Or is there reason to think it initiates the migration?

We did not see clear examples of centriole clusters in which the parental centriole was found in the lagging cluster, but there are technical challenges to being able to address this, and we cannot rule out this possibility. The challenge is due to the large distance between lagging and leading clusters and the inability to capture all of a single dendrite in the expanded images required for imaging the markers of centriole maturity. With respect to whether the mature centriole initiates migration, it is possible for a group that contains a mature centriole, although there is only one such centriole per cell so many migrating groups would not have a mature centriole.

5. To address concerns about the descriptive nature of the study, and provide some mechanistic insight into some aspect of this fascinating cell biology, the discussion mentions the possibility that Plk1 could control the timing of centriole maturation. Inhibitors of cell cycle kinases work well in explant cultures. The authors could test this hypothesis directly in explant culture, and I don't think it would be beyond the scope of the current study.

We agree with the reviewer that it would be interesting to test whether Plk1 inhibition could affect centriole maturation. We chose not to pursue these experiments at this time, in part because it would be difficult to validate the extent of Plk1 inhibition given that the effects of Plk1 inhibition are best observed in mitotic cells and there are relatively few such cells in the explants. But we do take the recommendation for future studies.

Reviewer #2 (Recommendations for the authors):1. The ALI culture is really impressive and important but I think needs a little more data to justify its usefulness. While the authors have shown nicely that there is still some proliferation (based on EdU), I am not sure that they have sufficiently shown that the whole migration process is actually happening. While this is somewhat inferred from the taxol data it is an important aspect to this experimental paradigm that should be more clearly shown.

We thank the reviewer for their comment and believe that this system can replicate the important early steps of centriole amplification and migration to the apical surface. By EdU incorporation, we show that DNA synthesis occurs in the explant system. As the reviewer notes, after perturbation of the system with taxol treatment, we show that early steps of migration are occurring in the system. Finally, using live imaging, we also show that late steps of migration, including incorporation into the apical surface, are occurring. We’ve added more text addressing these points in the manuscript.

2. The MT analysis seems very preliminary. "Our finding that microtubule nucleation proteins are present on migrating centrioles suggests another, non-mutually exclusive hypothesis: that migrating centrioles are microtubule organizing centers, with self-nucleated microtubules involved in their motility"….I think this is very likely and really should be addressed. I think a deeper analysis (see point 3) and additional drug treatments (eg nocodazole) are necessary. While genetic analysis of motors would be exciting and likely to yield interesting results it is appreciated that this is outside the scope of this manuscript.

We agree with the reviewer that it would be helpful to have more information about the microtubule requirement. Please see response to Essential Revision 1.

3. OSN dendrites have a polarized microtubule cytoskeleton, in which the microtubule minus ends are oriented out, toward the dendritic tips (Burton, 1985)…While Burtons work is certainly beautiful, I think the relevant question here is what is the orientation of MTs during migration. One might not expect (-) ends at the OSN dendrite until centrioles arrive. Do migrating centrioles nucleate MTs in an oriented manner? This seems critical to determine the role of MTs in centriole migration. Additionally, what does the MT network look like during this centriole migration. I think better imaging of MTs/(-) end markers and (+) end markers would help explain the role of MTs in this process better.

Please see response to Essential Revision 2.

4. Do the dendritic knobs position properly in the absence of centrioles / MTs. Perhaps this is known but it was unclear to me from the data and it seems important to know if the dendrites are positioned prior to centriole migration or do the first wave of centrioles act as leaders that push (or pull) the dendrite to its proper site (as is sort of drawn in the cartoon).

We agree that it would be interesting to further explore these possibilities, but have limited ability to do so given the available tools. We can say that the ends of growing dendrites can be far (as much as 8.4 μm in our experiment) from the closest group of centrioles, suggesting that the centrioles don’t “lead from the tip.” However, they could be providing the microtubules required for outgrowth and/or dendrite knob positioning. We did try to perturb microtubules with nocodazole treatment, but can make only limited conclusions because of the likely presence of stable microtubules in the dendrites, as described above. We hope in the future to be able to leverage tools to prevent centriole duplication, ideally specifically in developing OSNs, to directly test the contribution of amplified centrioles to the process.

5. "the subapical region, and the middle and basal region (regions marked in Figure 1A)" ….please explain how these were defined experimentally.

We thank the reviewer for pointing out this oversight on our part. We have now described this more fully in the text and in a new figure, Figure 5-Supplementary Figure 1A. See also the response to reviewer 1 above.

6. The quantifications need statistical analysis.

We now included statistical analysis for the taxol treatment. Using all categories for male and female samples, we performed a permutation test on the paired data (see R file referenced in Methods for details). Overall, we found the mean difference between paclitaxel treatment and the DMSO control condition to be 1.38 centriole groups per 100 μm with a p-value of 0.0084 when resampling 10,000 times.

Reviewer #3 (Recommendations for the authors):One of the most interesting aspects highlighted by the authors in this and their previous study is the atypical regulation of centriole assembly in olfactory neuron progenitors. However, some important aspects of these mechanisms remain unclear. As proposed by the authors in the discussion paragraph, it is possible that centrioles are also assembled in immature OSNs. The migrating centrioles would then belong to 3 different generations – a mother centriole, daughter centrioles formed in the previous cycle, and procentrioles formed in the OSNs. This could explain that the number of centrioles is slightly lower in immature OSNs than in mature ones (although this might result from experimental issues, as in Supplementary Figure 1A, centriole labeling appears sometimes incomplete and difficult to interpret), that only some of the migrating centrioles still possess SAS6 and STIL, and that migrating centrioles appear to vary in length (Supplementary Figure 1A, Figure 2A, 3B).It would be important to establish whether STIL and SAS-6 labeled centrioles can be formed at the previous cell cycle, or whether they are formed in immature OSNs. Is it possible to analyze immature OSNs in G1 phase? Alternatively, could they use centrinone to block centriole duplication in immature OSNs and determine if this leads to a loss of STIL/SAS6 signaling during migration?

The reviewer’s question is a good one – is the presence of STIL and SAS-6 indicative of the time in OSN differentiation when a given centriole formed? We believe that the most parsimonious explanation for the presence of only some centrioles bearing these markers is that the STIL/SAS-6 labeled centrioles are formed in immature OSNs after mitotic divisions and before (or during) dendrite outgrowth, i.e., they should all be in G1. This is based on both our previous publication about centriole amplification in OSNs (Ching and Stearns, 2020) and the current manuscript. We would very much like to inhibit centriole duplication to address this question, as described above in response to reviewer #2, but have yet to work out the methods to do so (for example, genetic methods, nasal delivery of centrinone, or long-term explant culture, etc.).

– The authors propose that there may be spatial control of centriole maturation (e.g., line 393). However, it is also possible that this control is simply temporal. In the first case, inhibition of dendrite extension should block centriole maturation. Is it possible to block dendrite extension and/or centriole migration, for example by using drugs that depolymerize actin or microtubules?Regarding the mechanisms underlying centriole migration, the observation that dendrite extend concomitantly to centriole migration is interesting but could be more substantiated. Ideally, this would involve specifically blocking one or the other, which is probably difficult to achieve in this context. Again, is it possible to use cytoskeleton drugs against actin or microtubules to do so? Alternatively, do leading centriole groups sometimes move backwards, and if so, does the dendrite continue extending?

We agree with the reviewer that the control of centriole maturation could be temporally regulated, for example as part of the differentiation program of OSNs, and have edited the manuscript to account for this possibility. With respect to uncoupling centriole migration and dendrite outgrowth, we note, as above, that nocodazole treatment is probably not able to depolymerize all microtubules in this system, and depolymerization of actin is likely to alter the overall tissue architecture, and thus was not attempted. In live imaging we do see examples of centriole groups moving backwards or remaining stationary during a period and it would be interesting to ask the fate of the dendrite tip in such cases, but we do not yet have the ability to co-label centrioles and the dendrite membrane for live imaging.

– The absence of rootletin labeling in centrioles is used as an argument to exclude a rootlet involvement in migration. However, the labeling is also rather weak in centrioles of mature OSNs. Is it known that the rootlet is poorly developed in these cells (e.g., as seen by EM), or is this due to labeling issues with the antibody used in this study? In this case, it is not clear that developing rootlets would be detected efficiently in migrating centrioles.

To our knowledge, our work represents the first description of ciliary rootlets in OSNs. As reviewed in McClintock et al., Chem Senses (2020), “Although OSNs have been shown to express components of the ciliary rootlet (Yamamoto 1976; McClintock et al. 2008), it is still unknown whether olfactory cilia have a rootlet.” Based on that characterization, which includes the absence of prominent rootlets as one sees in some other ciliated cell types, we believe that the “rather weak” labeling in OSNs probably does reflect the nature of rootlets in this tissue. The antibody that we used does robustly label rootletin in cohesion fibers in cycling cells grown in culture, and we have added a figure showing this (Figure 3- Supplementary Figure 1).